# Drug-resistant EGFR mutations promote lung cancer by stabilizing interfaces in ligand-free kinase-active EGFR oligomers

R. Sumanth Iyer [1,9,10], Sarah R. Needham [1,10], Ioannis Galdadas[2,3,10], Benjamin M. Davis[1,10], Selene K. Roberts[1,10], Rico C. H. Man[4], Laura C. Zanetti-Domingues [1], David T. Clarke[1], Gilbert O. Fruhwirth [4], Peter J. Parker [5,6], Daniel J. Rolfe [1] ✉, Francesco L. Gervasio [2,3,7,8] ✉ & Marisa L. Martin-Fernandez [1] ✉

The Epidermal Growth Factor Receptor (EGFR) is frequently found to be mutated in non-small cell lung cancer. Oncogenic EGFR has been successfully targeted by tyrosine kinase inhibitors, but acquired drug resistance eventually overcomes the efficacy of these treatments. Attempts to surmount this therapeutic challenge are hindered by a poor understanding of how and why cancer mutations specifically amplify ligand-independent EGFR auto-phosphorylation signals to enhance cell survival and how this amplification is related to ligand-dependent cell proliferation. Here we show that drug-resistant EGFR mutations manipulate the assembly of ligand-free, kinase-active oligomers to promote and stabilize the assembly of oligomer-obligate active dimer sub-units and circumvent the need for ligand binding. We reveal the structure and assembly mechanisms of these ligand-free, kinase-active oligomers, uncovering oncogenic functions for hitherto orphan transmembrane and kinase interfaces, and for the ectodomain tethered conformation of EGFR. Importantly, we find that the active dimer sub-units within ligand-free oligomers are the high affinity binding sites competent to bind physiological ligand concentrations and thus drive tumor growth, revealing a link with tumor proliferation. Our findings provide a framework for future drug discovery directed at tackling oncogenic EGFR mutations by disabling oligomer-assembling interactions.

Epidermal growth factor receptor (EGFR) is a transmembrane tyrosine kinase receptor at the heart of signals for cell survival, growth and division[1]. EGFR signals for growth and division are regulated on the cell surface by the binding of cognate growth factor ligands, which promote the assembly of a two-liganded back-to-back ectodomain dimer (B2B$^{ect}_{dimer}$)[2]. This dimer underpins across the membrane the formation of an asymmetric kinase dimer (Asym$^{kin}_{dimer}$)[3], which is fundamental to catalyze EGFR auto-phosphorylation in C-terminal tyrosine residues[1] (Fig. 1a). In the context of ligand-bound oligomers, EGFR auto-phosphorylation triggers EGFR-dependent canonical downstream signaling pathways, like PI3K/Akt and Ras-MAPK, which promote cell growth and proliferation[4–6].

Non-small cell lung cancer (NSCLC) has a 5-year relative survival rate of 28%[7,8]. Important examples of oncogenic drivers of NSCLC are

somatic EGFR mutations in exons 18-21, frequently identified (10-60% of lung adenocarcinomas[9–11]) among patients successfully treated with first-generation ATP-competing tyrosine kinase inhibitors (TKIs)[12], and in-frame insertions of three or more base pairs in exon 20 (Ex20Ins), unresponsive to TKIs and accounting for 4-10% of all EGFR mutations in NSCLC[13]. In TKI-responsive patients, resistance nevertheless emerges with high frequency[14–17] through the development of a dominant secondary T766M mutation in exon 20 at the entrance of the ATP-binding site (also known as T790M when the 24 amino acid signal peptide is included) (Fig. 1a), that increases ATP-binding affinity[18]. Partial sensitivity is maintained with second-generation irreversible TKIs[19], like Afatinib[20], that form covalent bonds with a cysteine residue in the ATP-binding pocket, but their potency against wild-type (WT)-EGFR induces severe epithelium-based toxicity[21]. This limitation was surmounted with highly selective, third-generation irreversible TKIs, such as Osimertinib[22,23]. However, initial efficacy is overcome by the acquisition of mutations in residues that form covalent bonds with irreversible inhibitors, like C773S[24–26]. Mutant-selective allosteric drugs (so-called fourth-generation) like EAI045[27] overcome resistance to third-generation TKIs by preventing the kinase domain from adopting its active conformation when combined with Cetuximab to also block ligand-induced EGFR dimerization[27–29]. However, toxicity-related concerns from off-target effects of Cetuximab limit therapeutic potential[30,31].

Current approaches target either the ATP-binding pocket (Fig. 1a), the conformation of the kinase monomer and/or ligand-induced dimerization. One yet untested strategy is to interfere with the poorly characterized ligand-independent (or constitutive) kinase-active state of EGFR, which elicits non-canonical, EGFR-dependent signals for cell survival[32,33]. This state is selectively amplified by NSCLC mutations, typically resulting in a few-fold increase in ligand-independent autophosphorylation[34], and could therefore underpin ligand-dependent tumor proliferation. However, the associated mechanisms remain poorly characterized. The best understood ligand-free structures have inactive kinases, namely EGFR monomers and auto-inhibited dimers made up of a ligand-free B2B$^{ect}_{dimer}$ structurally coupled to a symmetric head-to-head kinase dimer (B2B$^{ect}$/H2H$^{kin}_{dimer}$)[35] (Fig. 1b). Exploiting super-resolution Fluorophore Localization Imaging with Photobleaching (FLImP)[6,36–40], a molecular ruler that measures inter-receptor separations in specifically-labeled dimers and oligomers on cells (Fig. 2), guided by protein structures and molecular dynamics (MD) simulations, we previously made progress

towards understanding further the ligand-independent state of an autoinhibited oligomer EGFR architecture based on repeats of a head-to-head ectodomain dimer linked to non-interacting kinase monomers (H2H$^{ect}$/2x$^{kin}_{monomers}$)[36] (Fig. 1b). We also previously detected the FLImP-separation signatures of two full-length ligand-free dimer conformers. One is consistent with the structure of the B2B$^{ect}$/H2H$^{kin}_{dimer}$[35] (Fig. 1b). In the other, the Asym$^{kin}_{dimer}$ is structurally coupled to a ligand-free stalk-to-stalk ectodomain dimer (St2St$^{ect}_{dimer}$)[36] (Fig. 1b). However, in the St2St$^{ect}$/Asym$^{kin}_{dimer}$, which could account for ligand-independent signals, the Asym$^{kin}_{dimer}$ is disfavored by the WT-EGFR kinase[41,42]. Moreover, when the Asym$^{kin}_{dimer}$ is artificially joined to two ligand-free ectodomains, the dimer is disordered in electron micrographs[43,44]. Intriguingly, the T766M mutation, despite increasing EGFR autophosphorylation in the absence of ligand, counterintuitively destabilizes the catalytic Asym$^{kin}_{dimer}$ in favor of the inactive H2H$^{kin}_{dimer}$[36] (Fig. 1b). Given this, we conjectured that the St2St$^{ect}$/Asym$^{kin}_{dimer}$ might be an obligate oligomer sub-unit chaperoned by the B2B$^{ect}$/H2H$^{kin}_{dimer}$ in hetero$^{conf}$-oligomers. However, the previous ~5 nm resolution of FLImP prevented us from testing the stated hypothesis because we could not resolve some dimer separations from each other and/or from those that arise from the interaction between dimer sub-units in an oligomer.

Here we implement a higher-resolution FLImP version which, combined with large-scale simulations of various membrane-embedded dimer interfaces, allowed us to build an experimentally backed model of all the relevant interactions required to assemble St2St$^{ect}$/Asym$^{kin}_{dimer}$ sub-units within ligand-free EGFR oligomers. We show that WT-EGFR, T766M-EGFR, and Ex20Ins-EGFR share a ligand-free hetero$^{conf}$-oligomer structure in which scaffolds made of H2H$^{ect}$/2x$^{kin}_{monomers}$ and B2B$^{ect}$/H2H$^{kin}_{dimer}$ sub-units held by a transversal transmembrane interface[45] cantilever into position the extracellular portion of the St2St$^{ect}$/Asym$^{kin}_{dimer}$ under the regulation of the ectodomain tethered conformation[46]. Within these hetero$^{conf}$-oligomers, St2St$^{ect}$/Asym$^{kin}_{dimer}$ sub-units are positively and negatively regulated via two hitherto functionally-orphan kinase interfaces (PDB IDs:3VJO [https://www.rcsb.org/structure/3VJO][47] and in 5CNO [https://www.rcsb.org/structure/5CNO][48]). Stabilization by T766M and Ex20Ins NSCLC mutations of the ancillary kinase interfaces leads to an increase in the number and stability of St2St$^{ect}$/Asym$^{kin}_{dimer}$ sub-units, which accounts for mutation-dependent increases in ligand-free phosphorylation. Excitingly, our finding that St2St$^{ect}$/Asym$^{kin}_{dimer}$ sub-units are

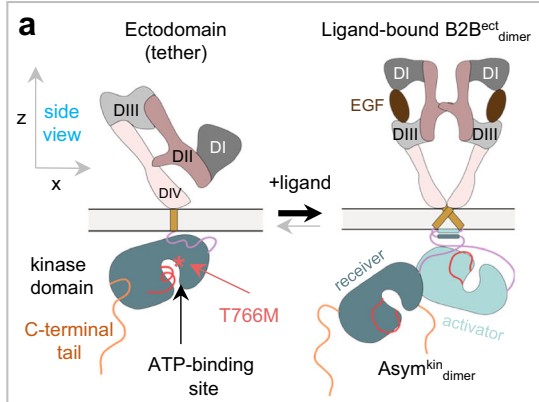
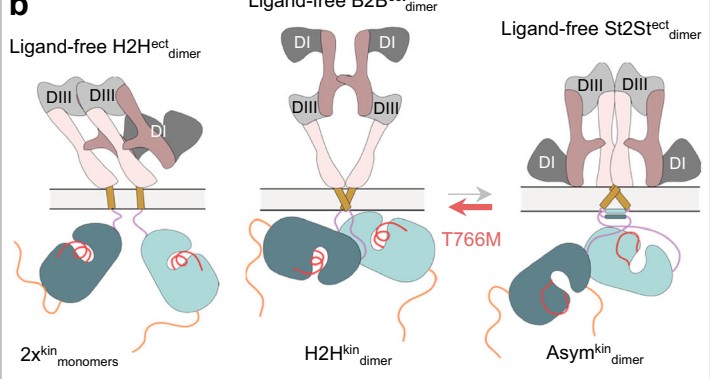

**Fig. 1 | Cartoon models of EGFR dimers. a** Left, Cartoon of a ligand-free tethered ectodomain (subdomains numbered) linked to a kinase monomer; the ATP-binding site, T766M mutation, and C-terminal tail are marked. Right, A two-liganded (EGF) extended back-to-back ectodomain dimer (B2B$^{ect}_{dimer}$) structurally coupled across the plasma membrane[80] to an asymmetric kinase dimer (Asym$^{kin}_{dimer}$)[3], in which an "activator" kinase (teal) stabilizes a "receiver" kinase (dark green) in the active

conformation. **b** Left, a ligand-free head-to-head ectodomain dimer (H2H$^{ect}_{dimer}$) sub-unit linked to two kinase monomers[36]. Middle, a ligand-free back-to-back ectodomain dimer (B2B$^{ect}_{dimer}$) sub-unit[81] coupled to a head-to-head kinase dimer (H2H$^{kin}_{dimer}$) sub-unit[35]. Right, a ligand-free stalk-to-stalk ectodomain dimer (St2St$^{ect}_{dimer}$) sub-unit[36] coupled to the Asym$^{kin}_{dimer}$ sub-unit[36].

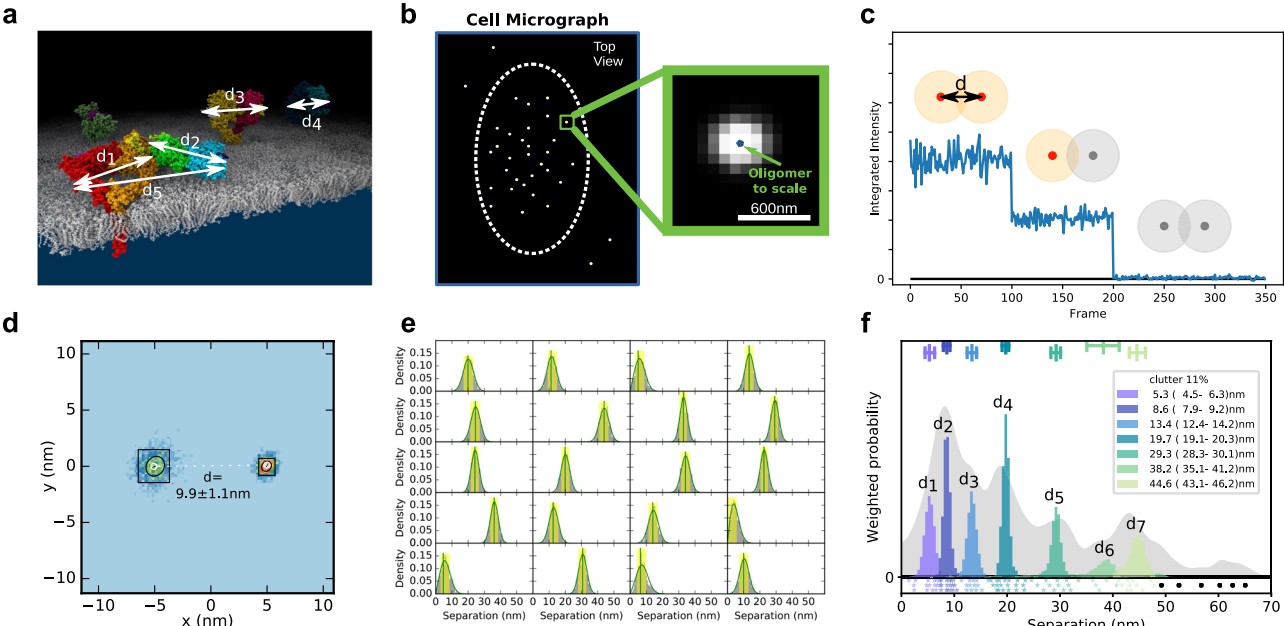

**Fig. 2 | Key stages in the FLImP data acquisition and analysis process. a** EGFR is a transmembrane protein that forms higher-order associations on the scale of 0-70 nm. EGFR is amenable to fluorescent labeling at several sites, in this case a specific site of the extracellular domain (DIII) (the intracellular domain is not depicted); the distances (d1-dn) provide a detailed structural signature. **b** When visualized under TIRF microscopy in cells, the size of labeled-EGFR dimers and oligomers is smaller than the diffraction limit of the microscope. Therefore, the fluorescent tags associated to EGFR dimers and oligomers emit light within a diffraction-limited point-spread function (PSF), which is also the microscope image of a single molecule (yellow spots). To resolve the positions of multiple EGFR molecules within a single diffraction limited spot, a video acquisition (FLImP raw data) is taken as the fluorophores photobleach. Single-molecule feature detection and tracking of these videos reveals integrated intensities through time and a subset of these spots which have multistep photobleaching (**c**) is identified (track selection); such single molecules bleach in single steps, and this can be used to estimate the number of fluorophores emitting light (red) in the spot as a function of time. By combining this information with prior knowledge of PSF shape, we can then fit the selected spots through time with a varying sum of Gaussian PSFs model to determine the positions of the emitting fluorophores (FLImP localization fit) with associated uncertainties (**d**). Multiple such measurements in the form of empirical posteriors (**e**) can be pooled (summed) into a FLImP signature of the structure (**f**, gray histogram). The green vertical line in **e** is the mean and the yellow shading represents a 70% confidence interval. Finally, a Bayesian decomposition of the separation measurement set (**e**) is performed to determine the most likely set of unique discrete separations present within the structure (**f**, colored components). In **f** the legend and bars above colored component distributions give the median and most-compact 68% confidence interval for each. Legend also gives median proportion of measurements assigned to clutter. (More in Supplementary Note 3).

epidermal growth factor (EGF) ligand high affinity binding sites, which are the sites previously proposed to drive tumor growth under pM physiological ligand concentrations[49], explains how the dysregulation of the ligand-free kinase active state by NSCLC mutations translates into their ability to potentiate EGFR-dependent tumor growth in vivo.

## Results

### FLImP reports oligomer size and sub-unit conformers

Photobleaching imaging correlation spectroscopy[50] analysis of CHO cells expressing ~$10^5$ copies/cell of WT-EGFR and EGFR mutations indicates that ~20–30% of EGFRs are incorporated in dimers and ~15–40% in oligomers (Supplementary Fig. 1a), consistent with previous results in fixed[36] and live cells[51]. FLImP measures the lateral pairwise separations on the scale of 0–70 nm between fluorescent probes specifically bound to the extracellular domain of protomers in non-monomer structures on the cell surface, and their relative abundance[6]. Mimicking the viewpoint of FLImP microscopy, we can assume the orthogonal projection onto the cell surface (xy-plane) of a hetero^conf-hexamer assembled by two dimer conformers (Fig. 3a). We created a dataset of synthetic FLImP separation probability distributions simulating measurements of individual separations between fluorophores bound to the hetero^conf-hexamer, including spurious fluorophore localizations (clutter), and noise, which we summed (Fig. 3b; gray background).

If individual dimeric units of different conformations are present on the cell surface, FLImP will report one separation per type. If, as shown in Fig. 3a, hetero^conf-oligomers are assembled by two dimer conformers, the FLImP peaks report 1st, 2nd, 3rd, etc., neighbor separations in the oligomer structure (Fig. 3b; color peaks). With the help of mutations, changes in intensity and position of these peaks can be used to investigate the structural sub-units assembling an oligomer. As an example, we simulated synthetic FLImP separations for the homo-trimer that forms when the dimer 1 conformer is disrupted (Fig. 3c, gray background). FLImP decomposes from these data the remaining 1st and 2nd-order vertical separations from the homo-trimer (Fig. 3c, color peaks), revealing which 1st order separation belongs to the disrupted interface and which higher-order peaks are dependent on hetero^conf-interactions. The mutations and treatments that we used to dissect the interactions assembling ligand-free oligomers are mapped-out in Fig. 3d.

FLImP samples a finite population of separations, and this introduces errors. We, therefore, used bootstrap-resampling[52] to estimate how this affects the decomposition. Figure 3e illustrates that bootstrap-resampling can capture changes caused by the transition from hetero^conf-hexamer to homo-trimer and evaluate significance above finite sampling errors. These synthetic results suggest a resolution of <3 nm, which was experimentally validated in cells with a known membrane protein structure (Supplementary Fig. 1b).

Given enough resolution and with the help of mutations/treatments, separation peaks from 1st order interfaces are typically amenable to be assigned, but assigning higher order peaks can be harder. We interrogate the effect of mutations on unassigned higher order peaks by folding them into a multidimensional scaling (MDS)

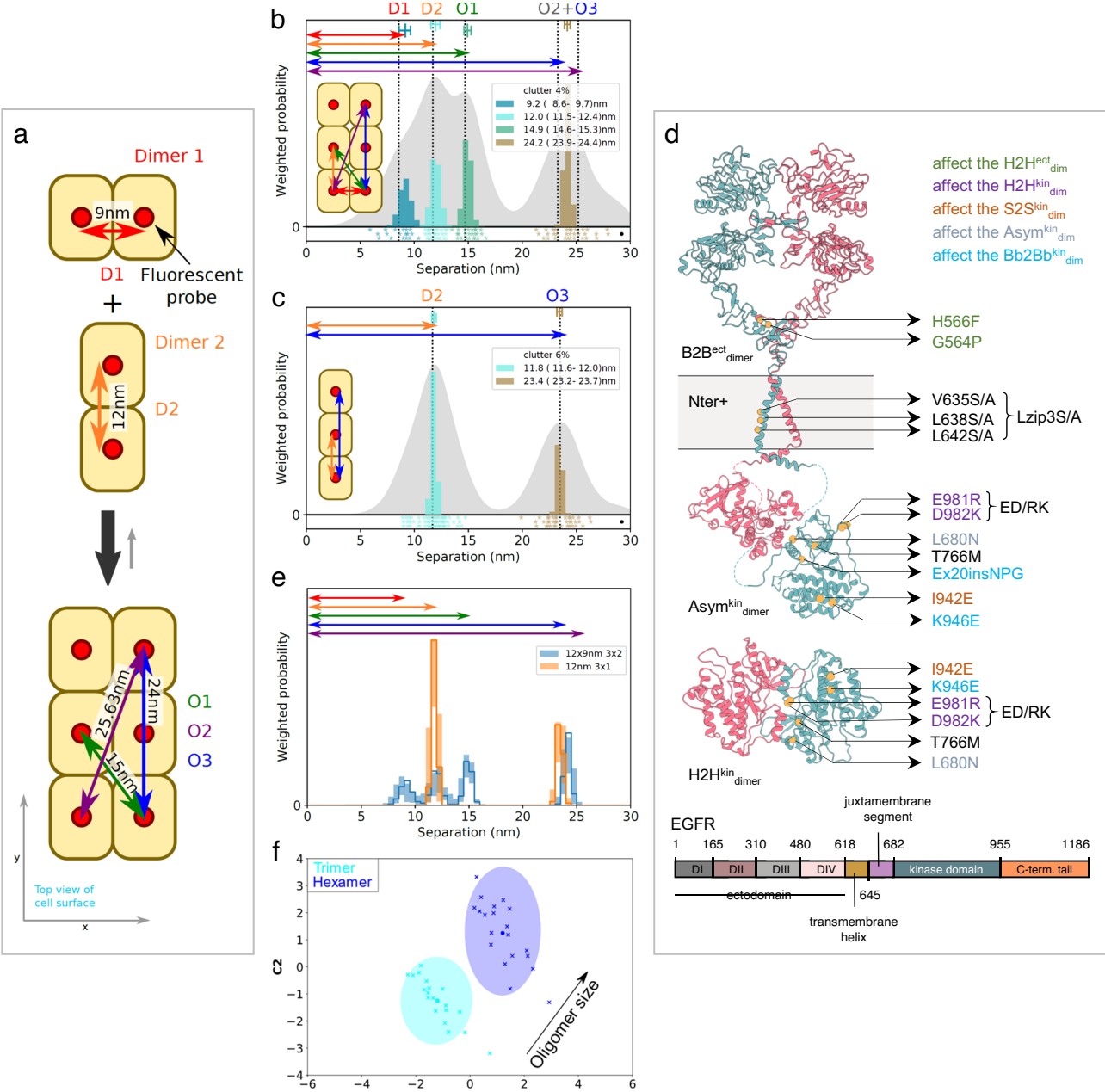

**Fig. 3 | FLImP measurement of pairwise separations. a** Cartoon dimer conformers assembling a hetero^conf-hexamer. Separations between bound fluorophores (red circles) are: two 1st-order interfaces (D1, D2), short and long diagonals (O1, O2), and 2nd-order vertical (O3). **b** FLImP analysis of 100 synthetic pairwise hetero^conf-hexamer separations (inset) (Supplementary Note 3). Sum of posteriors of individual separations between fluorophores (gray background) and abundance-weighted probability distributions of individual components of decomposed separation distribution (colored peaks). The area under each peak is weighted according to the estimated proportion of measurements attributed to that peak ("abundance"). Plot legend and bars above colored component distributions give the median and most-compact 68% confidence interval for each. Legend also gives median proportion of measurements assigned to clutter. Stars and dots below show individual separations assigned to peaks (colored) or clutter (black). **c** As (**b**) for the homo-trimer formed by the Dimer 2 conformer. **d** Top, map of mutations and treatments superimposed on the B2B^ect_dimer[81], an Asym^kin_dimer[3], and H2H^kin_dimer[54] sub-units. Mutations colored according to the different dimer conformers they inhibit or disrupt. Bottom, EGFR sequence diagram. **e** Comparisons between decomposed separation probability distributions between datasets. The continuous lines show the marginalized separation posterior for each condition, the sum of the abundance-weighted peaks in (**b** and **c**). Fluctuations around each continuous line arise from variations derived from FLImP decompositions for 20 bootstrap-resampled datasets to assess errors due to finite number of measurements. **f** Wasserstein MDS analysis of FLImP decompositions. This measures the work needed to convert a decomposed separation set into another, thereby estimating similarities and differences between whole FLImP separation decompositions. Similarities or dissimilarities between the 21 separation sets of different conditions (one main FLImP decomposition plus 20 bootstrap-resampled decompositions) are compared; in this case for the hetero^conf-hexamer (navy) (**b**) and homo-trimer (cyan) (**c**). The plot axes are components C1 and C2. C1 represents the dimension that captures the largest amount of data variance; C2 represents the second-largest amount of variance orthogonal to C1. The ellipse centers (95% confidence range) mark the positions of the main FLImP decompositions. Crosses mark the positions of individual bootstrap-resampled separation sets.

Wasserstein metric[53]. As shown in the example (Fig. 3f), we also include the bootstrap-estimated errors associated with finite sampling and calibrate changes to report oligomer growth direction.

## The B2B$^{ect}$/H2H$^{kin}_{dimer}$ and St2St$^{ect}$/Asym$^{kin}_{dimer}$ are oligomer sub-units

The color peaks in Fig. 4a show the most likely positions and intensities of pairwise separations between CF640R fluorophores specifically conjugated to anti-EGFR Affibodies bound to DIII of cell surface WT-EGFR ectodomains. All FLImP separation sets have hereafter the median positions of the WT-EGFR separations superimposed (dashed lines) to facilitate comparisons. FLImP measurements require immobilizing cell surface receptors by chemical fixation via a method demonstrated not to introduce detectable artefacts[6]. Nevertheless, predictions arising from the results of chemically fixed cells were validated in live cells using single particle tracking (Supplementary Fig. 1c), and by controls described below.

Interpreting a FLImP separation set requires assigning 1$^{st}$ order peaks to the structural sub-units assembling the underlying dimers/oligomers. In previous work[36], we linked the 10.8–13.5 nm peak (Fig. 4a, red dashed line) to the extracellular part of the B2B$^{ect}$/H2H$^{kin}_{dimer}$, which could also exist as an oligomer sub-unit (Fig. 4b). The link between the 10.8–13.5 nm peak and the B2B$^{ect}$/H2H$^{kin}_{dimer}$, either alone and/or as a protomer, is supported here by the observation that FLImP no longer indicates separation density between 8.1–14.8 nm when a previously proposed double E981R/D982K (ED/RK) mutation in the C-terminus[54] that destabilizes electrostatic interactions at the heart of the intracellular portion of the B2B$^{ect}$/H2H$^{kin}_{dimer}$ is introduced (Fig. 4c; red dashed line). By contrast, separation density at 9.2–13.2 nm increases when the H2H$^{kin}_{dimer}$ is stabilized by the T766M mutation[36] (Fig. 4d; red dashed line), further supporting the assignment of the 10.8-13.5 nm peak in the WT-EGFR separation set to the B2B$^{ect}$/H2H$^{kin}_{dimer}$ conformer.

We previously linked a <9 nm separation to the extracellular portion of the St2St$^{ect}$/Asym$^{kin}_{dimer}$[36], which could also exist as an oligomer sub-unit (Fig. 4b). Our enhanced FLImP method decomposes two <9 nm components in the WT-EGFR separation set (Fig. 4a; cyan and black dashed lines). A body of previous EGF-binding experiments suggested that EGFR is displayed on the cell surface in two forms, a minority (2–5%) of high-affinity EGFRs ($K_D$ = 10-100 pM) and a majority (95-98%) of low-affinity EGFRs ($K_D$ = 2–5 nM)[55]. Biophysical experiments suggested that these two affinity forms arise from two different dimer structures[56]. Given this, we speculated that if the two components of <9 nm arise from two different dimer conformers, we might be able to assign at least one according to ligand affinity. We pre-treated WT-EGFR-expressing cells with the conformation-selective monoclonal antibodies mAb-2E9[57] or mAb-108[58], which select for high and low-affinity EGF binding, respectively. As the selectivity of these mAbs is bona fide against EGF binding, mAb-treated cells were next probed with an EGF-CF640R derivative. Here cells were fixed after mAb treatment but before probing with EGF-CF640R to avoid ligand-induced conformational changes. Results show that EGF-CF640R binds well to fixed cells at similar sites to Affibody-CF640R (Supplementary Fig. 1d, e).

High-affinity EGFRs have been proposed to be mostly responsible for ligand-dependent EGFR's signaling[57]. Confocal results using mAb-2E9 indicate that EGF binds cell surface St2St$^{ect}$/Asym$^{kin}_{dimer}$ conformers with high affinity (Supplementary Fig. 1f). The FLImP separation set from mAb-2E9-treated cells suggests two components of <9 nm (4.9–6.8 nm and 7.2–8.9 nm) (Fig. 4e; cyan and black dashed lines), which are consistent with those in the WT-EGFR separation set. In contrast, no separation density at 6.8–9.9 nm is apparent in the set from cells treated with mAb-108, which blocks high-affinity binding (Fig. 4f; black dashed line). To assess the robustness of this result, because incompletely resolved components cannot be perfectly

separated, we compared the evidence for separations in bootstrap-resampled datasets after pooling the individual components (marginalized probability). This analysis indicates that the absence of separation density in the vicinity of ~8 nm associated to the mAb-108 treatment is robust to finite sampling errors (Fig. 4g). We therefore assigned the 7.0–8.6 nm peak in the WT-EGFR separation set to high-affinity EGF-binding St2St$^{ect}$/Asym$^{kin}_{dimer}$ sub-units. Further validation of this assignment is provided below.

Despite the T766M mutation destabilizing the Asym$^{kin}_{dimer}$[36], the separation density at ~8 nm is similar between WT-EGFR and T766M-EGFR (Fig. 4a, d). We hypothesized that the T766M mutation underpins the St2St$^{ect}$/Asym$^{kin}_{dimer}$ via interactions with the inactive B2B$^{ect}$/H2H$^{kin}_{dimer}$, which is stabilized by the T766M mutation[36]. To test this, WT-EGFR expressing cells were treated with Erlotinib[59], a TKI that binds to EGFR's kinase ATP-binding pocket, stabilizing the St2St$^{ect}$/Asym$^{kin}_{dimer}$[60]. With our previous poorer resolution, we found that Erlotinib treatment enhances a broad peak encompassing separations 1.6-10.2 nm[36]. Our higher resolution method decomposes three components under this peak (Fig. 4h). Interestingly, Erlotinib treatment recapitulates effects induced by the T766M mutation, most notably on separations of <20 nm associated with Erlotinib-treated WT-EGFR and T766M-EGFR (Figs. 4h, d). Quantified in Fig. 4i, both induce a decrease in the 5.8–7.1 nm component, the assignment and function of which is discussed below, and an increase at ~9–12 nm. These results argue that the stabilizing the St2St$^{ect}$/Asym$^{kin}_{dimer}$ directly by Erlotinib binding shares characteristics of stabilizing the B2B$^{ect}$/H2H$^{kin}_{dimer}$ via the T766M mutation, suggesting that the B2B$^{ect}$/H2H$^{kin}_{dimer}$ stabilizes the St2St$^{ect}$/Asym$^{kin}_{dimer}$ within hetero$^{conf}$-oligomers.

Further evidence that the B2B$^{ect}$/H2H$^{kin}_{dimer}$ and St2St$^{ect}$/Asym$^{kin}_{dimer}$ are structural sub-units in ligand-free hetero$^{conf}$-oligomers is shown in Fig. 4j. This includes the increase (decrease) in oligomer size induced by the T766M (ED/RK) mutations and the different oligomer sizes associated with mAb-2E9 and mAb-108. Notably, Erlotinib, as previously found[36], does not significantly increase oligomer size. These results assign the role of underpinning oligomer growth to the H2H$^{kin}_{dimer}$.

## H2H$^{ect}_{dimer}$/2x$^{kin}_{monomers}$ are sub-units in hetero$^{conf}$-oligomers

In a previous study[36], we proposed a third ligand-free dimer conformer based on a lattice contact in an X-ray structure of the tethered ectodomain monomer (PDB ID:4KRP [https://www.rcsb.org/structure/4KRP][61]), in which the monomers are held by ectodomain interactions (H2H$^{ect}_{dimer}$) (Fig. 5a). This model was supported by the FLImP results associated with ΔC-EGFR, a mutant in which the intracellular domains are deleted, here reanalyzed with the higher resolution decomposition (Fig. 5b). Results show peaks at almost a fixed interval consistent with the previously proposed homo-oligomers of repeating extracellular H2H$^{ect}_{dimer}$ units[36]. The presence of H2H$^{ect}_{dimer}$/2x$^{kin}_{monomers}$ is further supported by the finding that the 1$^{st}$-order separation of the truncated H2H$^{ect}_{dimer}$ of ΔC-EGFR shares position with the shortest component in the separation set for WT-EGFR (Fig. 5b; cyan dashed line). Taken together, previous and current evidence confirms the presence of H2H$^{ect}_{dimer}$/2x$^{kin}_{monomers}$ as a third ligand-free dimer conformer.

Higher resolution unmasked differences beyond the 1$^{st}$ order separation between the ΔC-EGFR and WT-EGFR sets (Fig. 5c). Given this, we speculated that H2H$^{ect}_{dimer}$/2x$^{kin}_{monomers}$ conformers might participate in hetero$^{conf}$-oligomerization. To investigate this, we introduced mutations that disrupt the H2H$^{ect}_{dimer}$. Our MD simulations suggest that inhibition of the tether via two well-understood DIV mutations in EGFR's ectodomain, H566F and G564P[55], disrupts the H2H$^{ect}_{dimer}$ but not the B2B$^{ect}_{dimer}$ (Supplementary Fig. 2 and Supplementary Tables 1 and 2). Consistent with previous data[55], our MD simulations also suggest that H566F, but not G564P, rearranges the DI of EGFR's ectodomain in a way that makes the binding site less

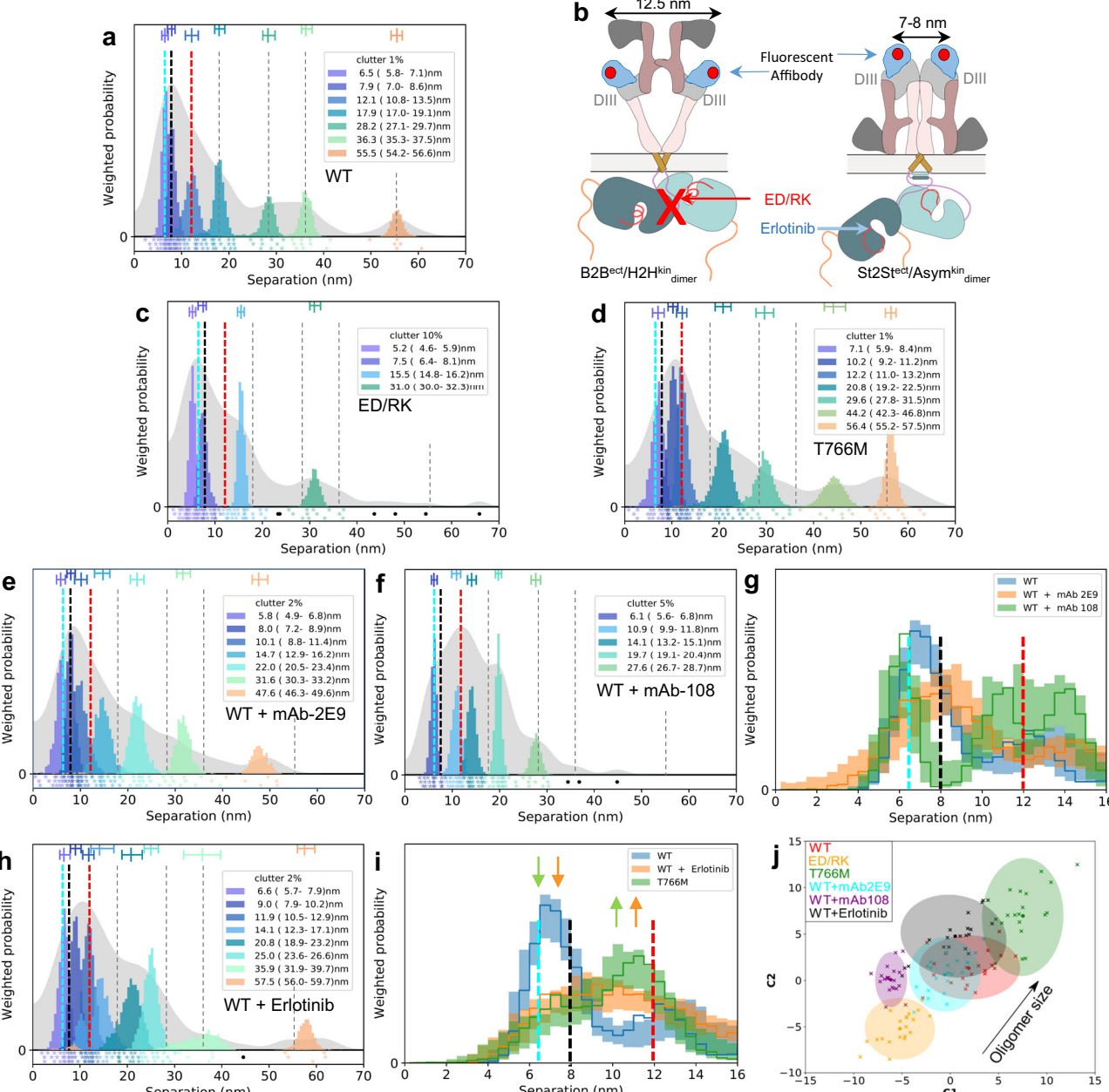

**Fig. 4 | The B2B$^{ect}$/H2H$^{kin}_{dimer}$ sub-unit underpins the formation of the St2St$^{ect}$/Asym$^{kin}_{dimer}$ sub-unit. a**, **c**–**f**, **h** FLImP analysis of 100 separation probability distributions between Affibody-CF640R pairs in the conditions indicated: Sum of posteriors of individual separations between fluorophores (gray background) and abundance-weighted probability distributions of individual components of decomposed separation distribution (colored peaks). Plot legend and bars above colored component distributions give the median and most-compact 68% confidence interval for each. Legend also gives median proportion of measurements assigned to clutter. Stars and dots below show individual separations assigned to peaks (colored) or clutter (black). The median peak positions marked by dashed lines are hereafter also superimposed on all the FLImP separation diagrams to facilitate comparisons with WT-EGFR. FLImP separations for all mutants and treatments are summarized in Supplementary Table 3. **b** Left, cartoon of a B2B$^{ect}$/H2H$^{kin}_{dimer}$ sub-unit labeled with two fluorescent anti-EGFR Affibody bound to the two DIII of the ectodomains. The ED/RK mutation inhibits the B2B$^{ect}$/H2H$^{kin}_{dimer}$ sub-unit. Right, a labeled St2St$^{ect}$/Asym$^{kin}_{dimer}$ sub-unit. Erlotinib binds the ATP

pocket of the kinase stabilizing the St2St$^{ect}$/Asym$^{kin}_{dimer}$ sub-unit. **g**, **i** Comparisons between decomposed separation probability distributions between FLImP datasets. The continuous lines show the marginalized separation posterior, i.e. the sum of the abundance-weighted peaks, for each condition in the inset. Fluctuations around each continuous line arise from variations derived from FLImP decompositions for 20 bootstrap-resampled datasets to assess errors due to the finite number of measurements. Median peak positions for WT-EGFR marked by dashed lines, as in **a**. **j** Wasserstein MDS analysis of FLImP decompositions for the conditions in the inset. Similarities or dissimilarities between the 21 separation sets of different conditions (one main FLImP decomposition plus 20 bootstrap-resampled decompositions) are compared. The plot axes are components C1 and C2. C1 represents the dimension that captures the largest amount of variance in the data, while C2 represents the second-largest amount of variance that is orthogonal to C1. The ellipse centers (95% confidence range) mark the positions of the main FLImP decompositions. The crosses mark the positions of individual bootstrap-resampled separation sets.

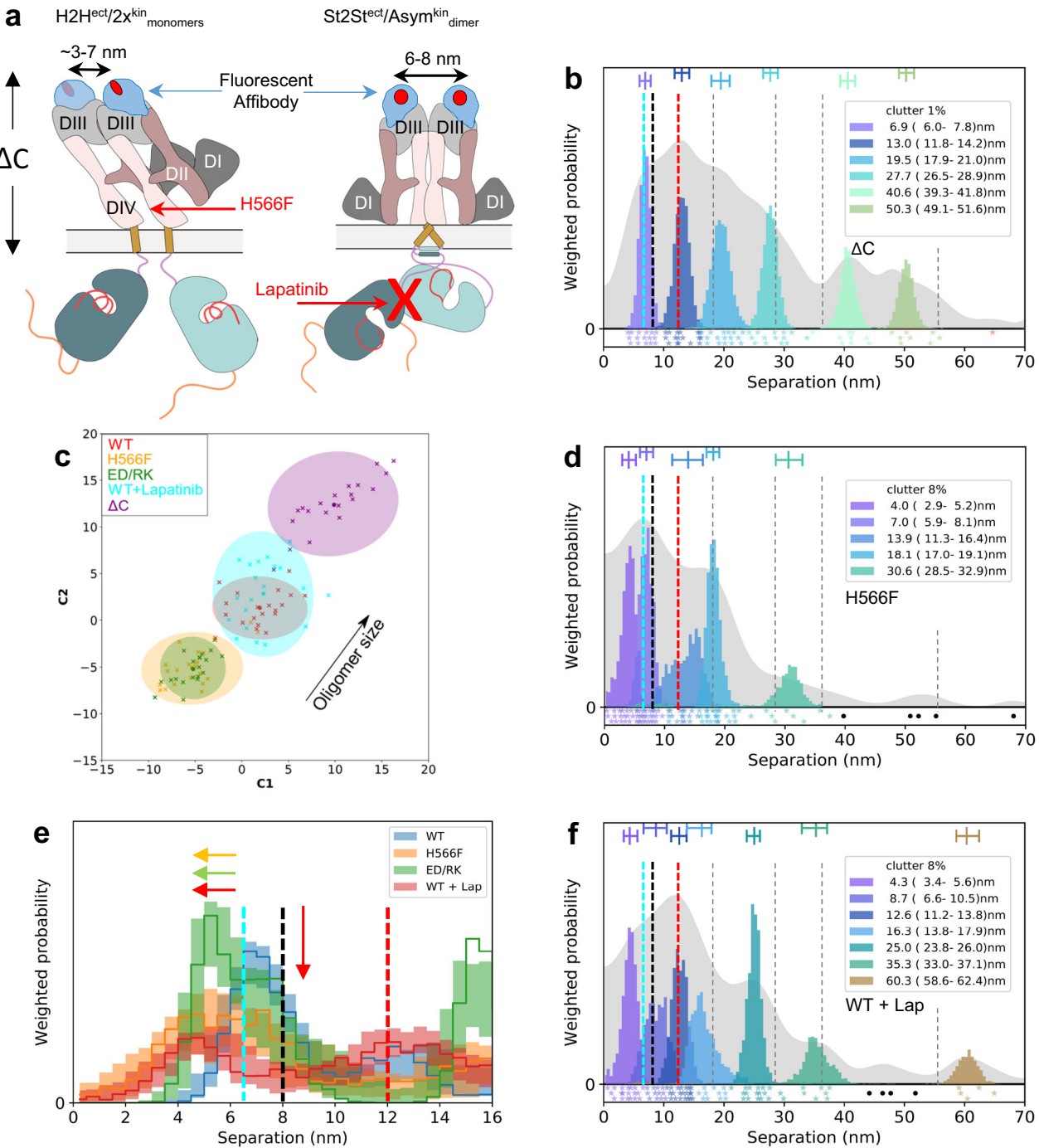

accessible (Supplementary Fig. 2). Using FLImP we found that H566F decreases oligomer size (Fig. 5c) and induces a shift in the shortest separation component assigned to H2H$^{ect}_{dimer}$/2x$^{kin}_{monomers}$ conformers from a median position of 6.5 nm to 4 nm, which implicates the ectodomain tethered conformation in the formation of H2H$^{ect}_{dimer}$/2x$^{kin}_{monomers}$ (Figs. 5d, e; blue and orange). Analogous results for G564P are in Supplementary Fig. 3a–d.

Because MD simulations suggest that the WT-EGFR H2H$^{ect}_{dimer}$ can explore a separation range of 3–7 nm between the center of mass of the two DIIIs (Supplementary Fig. 2c, Supplementary Note 1), the shift in the separation component annotated to H2H$^{ect}_{dimer}$/2x$^{kin}_{monomers}$ conformers from a median position of 6.5 nm to 5.2 nm introduced by the ED/RK mutation (Fig. 5e; blue and green) suggests that inhibiting the B2B$^{ect}$/H2H$^{kin}_{dimer}$ changes H2H$^{ect}_{dimer}$/2x$^{kin}_{monomers}$ conformation. The ED/RK mutations also induce a comparable oligomer size

reduction to that of the H566F mutation (Fig. 5c). Together these results suggest that B2B$^{ect}$/H2H$^{kin}_{dimer}$ sub-units interact with H2H$^{ect}_{dimer}$/2x$^{kin}_{monomers}$, and hence that the latter is also a sub-unit in ligand-free hetero$^{conf}$-oligomers.

From this, it follows that H2H$^{ect}_{dimer}$/2x$^{kin}_{monomers}$ might interact with the St2St$^{ect}$/Asym$^{kin}_{dimer}$. To test this, WT-EGFR expressing cells were treated with Lapatinib, a TKI that binds to EGFR's kinase ATP-binding site breaking the Asym$^{kin}_{dimer}$[62]. As expected, Lapatinib induced a significant reduction around the 8 nm position assigned to the St2St$^{ect}$/Asym$^{kin}_{dimer}$ conformer (Fig. 5e; blue and red, and 5f)[60]. This is also consistent with our finding that St2St$^{ect}$/Asym$^{kin}_{dimer}$ sub-units display high-affinity because Lapatinib additionally decreases EGF-binding affinity[60]. The reduction in the ~8 nm component is accompanied by a shift in the component assigned to the H2H$^{ect}_{dimer}$/2x$^{kin}_{monomers}$ conformer, from a median position of 6.5 nm to 4.2 nm

**Fig. 5 | The $\text{H2H}^{ect}_{dimer}/2x^{kin}_{monomers}$ sub-unit participates in $\text{hetero}^{conf}$-oligomer assembly. a** Left, cartoon of full-length $\text{H2H}^{ect}/2x^{kin}_{monomers}$ sub-unit with two fluorescent anti-EGFR Affibodies (blue with red spot) bound to ectodomain's DIIIs. The $\text{H2H}^{ect}_{dimer}$ is linked to two non-interacting kinase monomers[36]. ΔC-EGFR deletion mutant length is marked. DIV-binding tether-disruptive mutations (e.g. H566F) interfere with the $\text{H2H}^{ect}_{dimer}$ conformation. Right, a labeled $\text{St2St}^{ect}/\text{Asym}^{kin}_{dimer}$ sub-unit. Lapatinib binds the kinase ATP-binding pocket disrupting the $\text{Asym}^{kin}_{dimer}$. **b, d, f** FLImP analysis of 100 separation probability distributions between Affibody-CF640R pairs in the conditions indicated: Sum of posteriors of individual separations between fluorophores (gray background) and abundance-weighted probability distributions of individual components of decomposed separation distribution (colored peaks). Plot legend and bars above colored component distributions give the median and most-compact 68% confidence interval for each. Legend also gives median proportion of measurements assigned to clutter. Stars and dots below show individual separations assigned to peaks (colored) or clutter (black). WT-EGFR median peak positions marked by dashed lines. **c** Wasserstein MDS analysis of FLImP decompositions for conditions in the inset. Similarities or dissimilarities between the 21 separation sets of different conditions (one main FLImP decomposition plus 20 bootstrap-resampled decompositions) are compared. The axes are components C1 and C2. C1 represents the dimension that captures the largest amount of data variance, while C2 represents the second-largest amount of variance that is orthogonal to C1. The ellipse centers (95% confidence range) mark the positions of the main FLImP decompositions. The crosses mark the positions of individual bootstrap-resampled separation sets. **e** Comparisons between decomposed separation probability distributions between FLImP datasets. The continuous lines show the marginalized separation posterior, i.e. the sum of the abundance-weighted peaks, for each condition in the inset. Fluctuations around each continuous line arise from variations derived from FLImP decompositions for 20 bootstrap-resampled datasets to assess errors due to finite number of measurements. Note that the 8 nm separation corresponding to the $\text{St2St}^{ect}/\text{Asym}^{kin}_{dimer}$ sub-unit is not significantly decreased by the H566F mutation (orange), consistent with the tether-disrupting mutations not inhibiting phosphorylation[55] (Supplementary Fig. 3e). Dashed lines mark WT-EGFR median peak positions. Colored arrows show shift from WT.

(Fig. 5e, f; cyan dashed line), arguing that inhibiting the $\text{St2St}^{ect}/\text{Asym}^{kin}_{dimer}$ changes the conformation of $\text{H2H}^{ect}_{dimer}/2x^{kin}_{monomers}$, and hence that $\text{H2H}^{ect}_{dimer}/2x^{kin}_{monomers}$ and $\text{St2St}^{ect}/\text{Asym}^{kin}_{dimer}$ sub-units interact. Interestingly, Lapatinib does not decrease oligomer size (Fig. 5c). This will be discussed later with more data (Fig. 9).

### Ligand-free conformers interact via transmembrane contacts

The conformation of the three ligand-free dimer sub-units suggest $\text{hetero}^{conf}$-oligomer assembly might be mediated by transmembrane interactions. Therefore, we next considered the Lzip transmembrane dimer named after its leucine zipper-like interactions (Fig. 6a)[45]. In principle, the Lzip interface could mediate interactions between transmembrane monomers and dimers (Fig. 6b). To investigate this possibility, we mutated all three Lzip dimer transmembrane helix amino acids (V635, L638, L642) to either serine or alanine, named Lzip3S and Lzip3A, respectively. Based on previous literature, Lzip3S mutations would be expected to strongly inhibit the Lzip interaction, unlike the more conservative Lzip3A[63].

We found that short separations become poorly resolved in the Lzip3S-EGFR set (Fig. 6c). Notable effects include increased separation density at 2.2-4.6 nm (Fig. 6c, d), suggesting the conformation of $\text{H2H}^{ect}_{dimer}/2x^{kin}_{monomer}$ sub-units has changed, alongside an oligomer size reduction (Fig. 6e). The Lzip3A mutations did not show these effects (Supplementary Fig. 3g–i). Together, these results implicate Lzip contacts in $\text{hetero}^{conf}$-oligomer assembly.

To evaluate the consistency of the FLImP results, we reasoned that simultaneously inhibiting the $\text{H2H}^{kin}_{dimer}$ and $\text{Asym}^{kin}_{dimer}$, and thus all intra-dimer intracellular interactions, together with Lzip contacts should recapitulate the ΔC-EGFR results. This hypothesis was evaluated in two stages. First, by combining the ED/RK mutations with L680N, a kinase N-lobe mutation that inhibits the kinase domain from acting as receiver[54], and thus the $\text{St2St}^{ect}/\text{Asym}^{kin}_{dimer}$. Then the Lzip3S mutations were added. Reassuringly, the ED/RK + L680N mutations induce a shift in the peak assigned to $\text{H2H}^{ect}_{dimer}/2x^{kin}_{monomers}$ sub-units comparable to the Lzip3S mutations alone, confirming that the conformation of the $\text{H2H}^{ect}_{dimer}/2x^{kin}_{monomers}$ sub-units depends on $\text{hetero}^{conf}$-oligomer interactions (Fig. 6d, f). Also reassuringly, combining ED/RK + L680N + Lzip3S mutations recapitulated the results for ΔC-EGFR (Fig. 6e, g), e.g., pseudo-periodic separations, oligomer size increase, and the loss of the 2.8–4.3 nm peak, suggesting that Lzip contacts inhibit the formation of the homo-oligomers of repeating extracellular $\text{H2H}^{ect}_{dimer}$ interfaces.

### Extracellular structure of ligand-free $\text{hetero}^{conf}$-oligomers

Based on the above data, we constructed a model of the orthogonal projection on the cell surface of the $\text{hetero}^{conf}$-oligomer based on the known shape and dimensions of the three ligand-free dimer sub-units

(Fig. 1b, Supplementary Fig. 4). $\text{H2H}^{ect}_{dimer}/2x^{kin}_{monomers}$ and $\text{B2B}^{ect}/\text{H2H}^{kin}_{dimer}$ could form a tetramer in which, remarkably, the Lzip interface aligns one ectodomain of the $\text{H2H}^{ect}_{dimer}/2x^{kin}_{monomers}$ so it can link with another tetramer and form the extracellular portion of the $\text{St2St}^{ect}/\text{Asym}^{kin}_{dimer}$ (Fig. 7a).

To test this model and its relevance to the dysregulated ligand-free state, we implemented a 2D version of FLImP that reports triangular arrangements between probes bound to EGFR structures (Supplementary Fig. 5). The 3-fold higher probe concentration required for 2D FLImP was better suited to the less sticky EGF-CF640R derivative, as Affibody-CF640R at this higher concentration began to show signs of non-specific binding on the glass supporting the cells[64].

As an example, we probed cells expressing T766M-EGFR with EGF-CF640R after chemical fixation to avoid ligand-induced conformational changes, as discussed above. The resulting 2D FLImP triangle dataset was optimally grouped into seven distinct triangles (Fig. 7b–h). Reassuringly, the 1D separations in the triangles are found as components decomposed by 1D FLImP for T766M-EGFR and WT-EGFR (Fig. 4d, a), arguing that T766M-EGFR and WT-EGFR share $\text{hetero}^{conf}$-oligomer structure.

To evaluate whether the 2D FLImP data supports the proposed model, the model was expanded to the size required by the triangles and the positions expected for EGF-CF640R bound to EGFR's ectodomain DIII marked. We found that the smallest triangle (T1) accounts, within errors, for four triangular probe motifs in the model (Fig. 7i).Triangles T2-T7 each account for one motif (Fig. 7j). To further validate the applicability of the $\text{hetero}^{conf}$-oligomer structure to T766M-EGFR, we verified that combining the T766M mutation with the tether-disrupting H566F mutation or the Lzip3S mutations also disrupts the $\text{hetero}^{conf}$-oligomers (Supplementary Fig. 6). This, together with the excellent results of superimposing the triangles from 2D FLImP data collected from CHO cells expressing T766M-EGFR indicate that the model is an accurate representation of the extracellular portion of the ligand-free $\text{hetero}^{conf}$-oligomers. Interestingly, this model predicts that the smallest oligomers that could bear one $\text{St2St}^{ect}/\text{Asym}^{kin}_{dimer}$ sub-unit are a hexamer assembled from two $\text{H2H}^{ect}_{dimer}/2x^{kin}_{monomers}$ sub-units and one $\text{B2B}^{ect}/\text{H2H}^{kin}_{dimer}$ and an octamer assembled from two of each (Fig. 7a). Consistent with this, mutations that preserve phosphorylation are at least hexamer in size (Supplementary Table 3).

### A ligand-independent mechanism of Ex20ins-induced activation

The model predicts that inhibiting the $\text{B2B}^{ect}/\text{H2H}^{kin}_{dimer}$ conformer would result in the tetramer shown in Fig. 8a. Inhibiting the Lzip interface should have the same effect, but separations found when $\text{B2B}^{ect}/\text{H2H}^{kin}_{dimer}$ sub-units are inhibited by the ED/RK mutations are inconsistent with those found when Lzip contacts are inhibited via

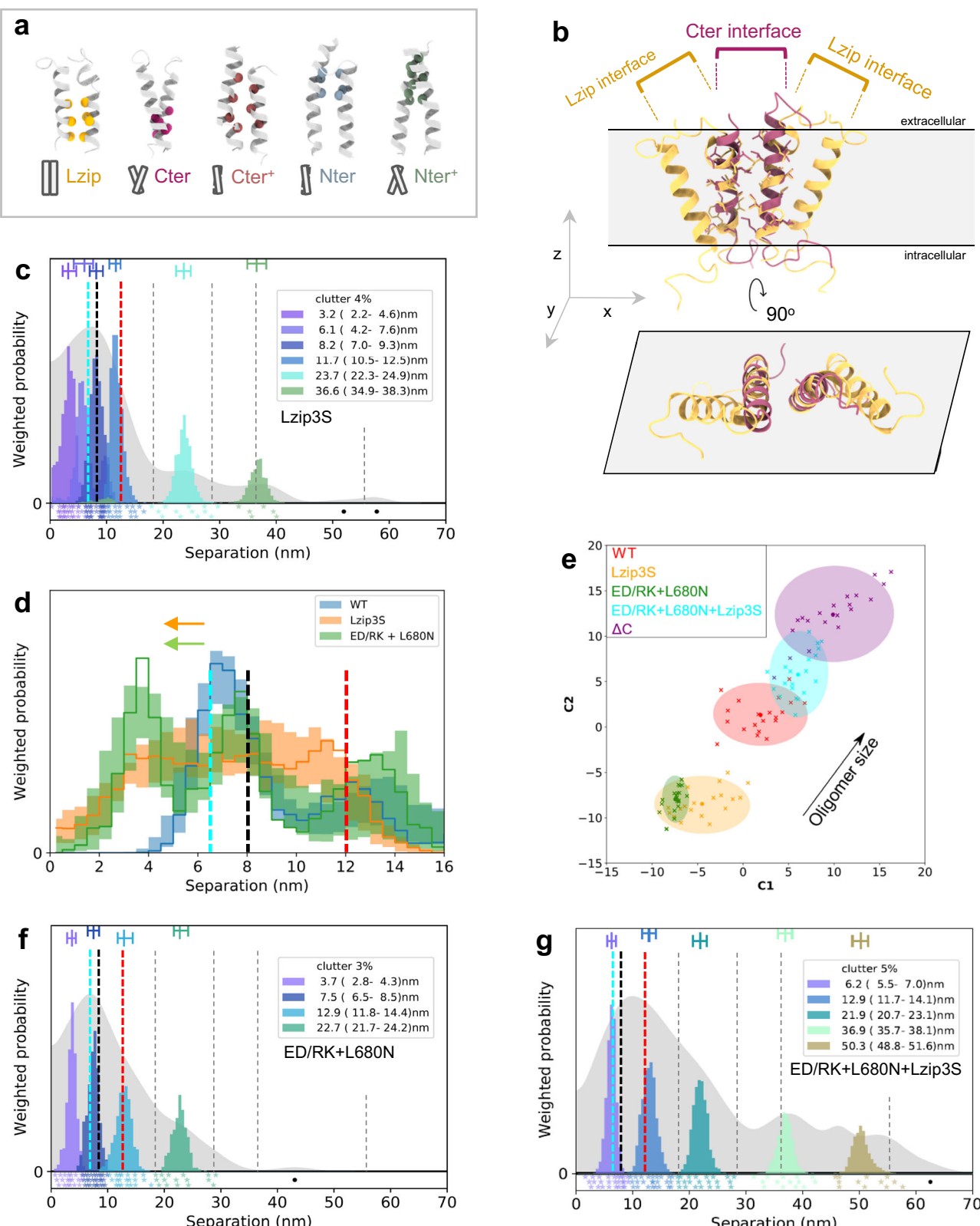

Lzip3S mutations (Fig. 8b). This hinted at the possibility that we had not yet accounted for all hetero[conf]-oligomer assembling interactions.

We considered a functionally orphan symmetric backbone-to-backbone kinase interface (Bb2Bb[kin]_interface), revealed by X-ray crystallography[47] (Fig. 8c). Because a kinase monomer could dock into a Bb2Bb[kin]_dimer to form an Asym[kin]_dimer, we speculated that the Bb2Bb[kin]_interface might be involved in strengthening the St2St[ect]/

Asym[kin]_dimer sub-units. Giving credence to this possibility, two non-naturally occurring charge-reversal R938E and K946E mutations, which compromise the Bb2Bb[kin]_dimer, as reported by MD simulations (Supplementary Fig. 7a), decrease receptor phosphorylation (Fig. 8d).

Remarkably, we found that the separation set associated with K946E-EGFR is almost indistinguishable from that of ED/RK+L680N-EGFR (Fig. 8e–g). This reveals that disrupting the Bb2Bb[kin]_interface via

**Fig. 6 | Ligand-free dimer conformers interact via transversal transmembrane contacts. a** Previously proposed transmembrane dimers[45]. Lzip has leucine zipper-like interactions ($V^{635}$xxL$^{638}$xxxL$^{642}$) and could establish contacts with Nter$^+$/Nter and Cter$^+$/Cter to assemble oligomers. **b** Top, a speculative tetramer (yellow) from two monomers interacting through Lzip contacts with one Cter interface. A Cter dimer model (magenta) is placed on top of the tetramer showing that the residues in the Cter interface are distinct from Lzip. Bottom, orthogonal projection on xy-plane. (Equivalent for Lzip contacts with one Nter interface in Supplementary Fig. 3f). **c, f, g** FLImP analysis of 100 separation probability distributions between Affibody-CF640R pairs in the conditions indicated: Sum of posteriors of individual separations between fluorophores (gray background) and abundance-weighted probability distributions of individual components of decomposed separation distribution (colored peaks). Plot legend and bars above colored component distributions give the median and most-compact 68% confidence interval for each. Legend also gives median proportion of measurements assigned to clutter. Stars and dots below show individual separations assigned to peaks (colored) or clutter

(black). The median peak positions marked by dashed lines are those of WT-EGFR. **d** Comparisons between decomposed separation probability distributions between FLImP datasets. The continuous lines show the marginalized separation posterior, i.e. the sum of the abundance-weighted peaks, for each condition in the inset. The fluctuations around each continuous line arise from variations derived from FLImP decompositions for 20 bootstrap-resampled datasets to assess errors due to the finite number of measurements. Dashed lines mark WT-EGFR median peak positions. Colored arrows show shift from WT. **e** Wasserstein MDS analysis of FLImP decompositions for the conditions in the inset. Similarities or dissimilarities between the 21 separation sets of different conditions (one main FLImP decomposition plus 20 bootstrap-resampled decompositions) are compared. Plot axes are components C1 and C2. C1 represents the dimension that captures the largest amount of variance in the data, while C2 represents the second-largest amount of variance that is orthogonal to C1. The ellipse centers (95% confidence range) mark the positions of the main FLImP decompositions. The crosses mark the positions of individual bootstrap-resampled separation sets.

the K946E mutation recapitulates the effect of jointly inhibiting St2St$^{ect}$/Asym$^{kin}_{dimer}$ and B2B$^{ect}$/H2H$^{kin}_{dimer}$ sub-units via the combined ED/RK + L680N mutations. This is consistent with St2St$^{ect}$/Asym$^{kin}_{dimer}$ and B2B$^{ect}$/H2H$^{kin}_{dimer}$ conformers being obligate hetero$^{conf}$-oligomers. Adding the K946E mutation to T766M-EGFR also reduces oligomer size and decreases receptor phosphorylation (Fig. 8d, g, h, Supplementary Fig. 7c). Together, results suggest that the Bb2Bb$^{kin}_{interface}$ reinforces the St2St$^{ect}$/Asym$^{kin}_{dimer}$ sub-units, and thereby their interaction with the B2B$^{ect}$/H2H$^{kin}_{dimer}$ in the hetero$^{conf}$-oligomers.

If the above interpretation is correct, one would expect that stabilizing the Bb2Bb$^{kin}_{interface}$ should mirror the effects of stabilizing the St2St$^{ect}$/Asym$^{kin}_{dimer}$ conformer. We conjectured that Ex20Ins might increase the number of contacts between the kinase domains of the Bb2Bb$^{kin}_{interface}$, stabilizing that interface (Fig. 8i). This was supported by MD simulations of the WT and D770-N771insNPG (insNPG), a mutant chosen because structural data is available[65] (Supplementary Fig. 8a, b). In support of the notion that the Bb2Bb$^{kin}_{interface}$ underpins the stability of St2St$^{ect}$/Asym$^{kin}_{dimer}$ conformers, the separation set for insNPG-EGFR recapitulates the effects of the Erlotinib treatment in WT-EGFR (Fig. 8j, k, Supplementary Fig. 8c), revealing that the mechanism by which insNPG-EGFR dysregulates ligand-independent phosphorylation is by strengthening the Bb2Bb$^{kin}_{interface}$ in hetero$^{conf}$-oligomers.

### A ligand-independent mechanism of T766M-induced activation

After incorporating the Bb2Bb$^{kin}_{interface}$ (Fig. 9a), another prediction is that disrupting Lzip contacts should have an analogous effect on phosphorylation to inhibiting the B2B$^{ect}$/H2H$^{kin}_{dimer}$ via the ED/RK mutations, but only the latter increases phosphorylation (Fig. 8d). We speculated this might be explained if the B2B$^{ect}$/H2H$^{kin}_{dimer}$ sequestered kinase monomers from stabilizing the Asym$^{kin}_{dimer}$ via the Bb2Bb$^{kin}_{interface}$ (Fig. 9b). In the crystal lattice of the activator-impaired V924R H2H$^{kin}_{dimer}$[48] we noticed a side-to-side interface (S2S$^{kin}_{interface}$) that could play such a role (Fig. 9c).

We previously reported that the non-naturally occurring I942E mutation inhibits ligand-bound oligomerisation[36]. Here we show that I942E decreases ligand-free EGFR phosphorylation (Fig. 8d). The I942 residue lies at the heart of the S2S$^{kin}_{interface}$, and modelling of the WT and I942E mutant suggested that the latter stabilizes the S2S$^{kin}_{interface}$ (Supplementary Fig. 9a, Supplementary Note 1). The separation set associated with I942E-EGFR is consistent with the results obtained for WT-EGFR-expressing cells treated with Lapatinib, which inhibits the Asym$^{kin}_{dimer}$ (Fig. 9d, e). These results argue that S2S$^{kin}_{interface}$ stabilization via the I942E mutation also inhibits the St2St$^{ect}$/Asym$^{kin}_{dimer}$, indicating that the S2S$^{kin}_{interface}$ plays an autoinhibitory role. Combining the T766M and I942E mutations support this notion (Supplementary Fig. 9b, c).

If the B2B$^{ect}$/H2H$^{kin}_{dimer}$ sequestered kinase monomers via the S2S$^{kin}_{interface}$ preventing these from reinforcing St2St$^{ect}$/Asym$^{kin}_{dimer}$

sub-units via the Bb2Bb$^{kin}_{interface}$, then inhibiting the Bb2Bb$^{kin}_{interface}$ should make more kinase monomers available to stabilize the B2B$^{ect}$/H2H$^{kin}_{dimer}$ via the S2S$^{kin}_{interface}$, thereby growing larger oligomers as found when B2B$^{ect}$/H2H$^{kin}_{dimer}$ sub-units are stabilized by the T766M mutation (Fig. 4j). To test this possibility, we need to inhibit the Bb2Bb$^{kin}_{interface}$ whilst preserving St2St$^{ect}$/Asym$^{kin}_{dimer}$ and B2B$^{ect}$/H2H$^{kin}_{dimer}$ sub-units so they can compete for kinase monomers. However, when the Bb2Bb$^{kin}_{interface}$-inhibitory K946E mutation is introduced in WT-EGFR, the mutation also disassembles the St2St$^{ect}$/Asym$^{kin}_{dimer}$ and B2B$^{ect}$/H2H$^{kin}_{dimer}$ conformers (Fig. 8f). In contrast, partially disassembled Lzip3S-EGFR oligomers preserve their phosphorylation (Fig. 8d) and some density around 12 nm (Fig. 8b), suggesting that Lzip3S-EGFR oligomers retain some St2St$^{ect}$/Asym$^{kin}_{dimer}$ and B2B$^{ect}$/H2H$^{kin}_{dimer}$ sub-units. Thus, we reasoned that adding K946E to Lzip3S mutations should stabilize the B2B$^{ect}$/H2H$^{kin}_{dimer}$ and increase oligomer size. The results are consistent with this notion (Fig. 9f). We also found that the separation sets of Lzip3S + K946E and Lzip3S + L680N are similar below ~35 nm (Fig. 9g, h), suggesting that inhibiting the St2St$^{ect}$/Asym$^{kin}_{dimer}$ and the Bb2Bb$^{kin}_{interface}$ are almost equivalent, and explaining why disrupting the St2St$^{ect}$/Asym$^{kin}_{dimer}$, either directly via Lapatinib or indirectly via the I942E mutation, does not decrease oligomer size, as both stabilize the S2S$^{kin}_{interface}$ and thereby the H2H$^{kin}_{dimer}$ counterbalancing the effect of disrupting St2St$^{ect}$/Asym$^{kin}_{dimer}$ sub-units (Fig. 9f).

Results, therefore, show that destabilizing the St2St$^{ect}$/Asym$^{kin}_{dimer}$, either directly by the L680N mutation or indirectly by inhibiting the Bb2Bb$^{kin}_{interface}$, increases oligomer size. In WT-EGFR this depends on the S2S$^{kin}_{interface}$ at the expense of phosphorylation. The T766M mutation directly stabilizes the B2B$^{ect}$/H2H$^{kin}_{dimer}$, thus promoting the formation of larger and more stable[36] oligomers that can bear more St2St$^{ect}$/Asym$^{kin}_{dimer}$ sub-units, thus accounting for the increase in T766M-induced ligand-independent phosphorylation (Supplementary Fig. 7c).

### Tumor growth depends on hetero$^{conf}$-oligomerisation

To test the relevance of the proposed ligand-free oligomer structure and assembling mechanisms in vivo, we carried out cellular transformation assays using the IL3-dependent murine lymphoid Ba/F3 cell system with the aim of using these cells to establish tumor xenografts. Ba/F3 cells fail to survive and multiply in the absence of IL3[66,67], but this phenotype can be rescued by the ectopic expression of a constitutively active receptor tyrosine kinase, like for example T766M-EGFR, which allows survival signaling in the absence of IL3[66].

We generated Ba/F3 cell lines stably expressing WT-EGFR (Ba/F3 + WT), T766M-EGFR (Ba/F3 + T766M), and EGFR mutants at near equal levels using the PiggyBac system and tested the ability of the transformed cells to grow in the absence of IL3 (Supplementary Fig. 10a). To minimize mice number, from the mutations that disrupt

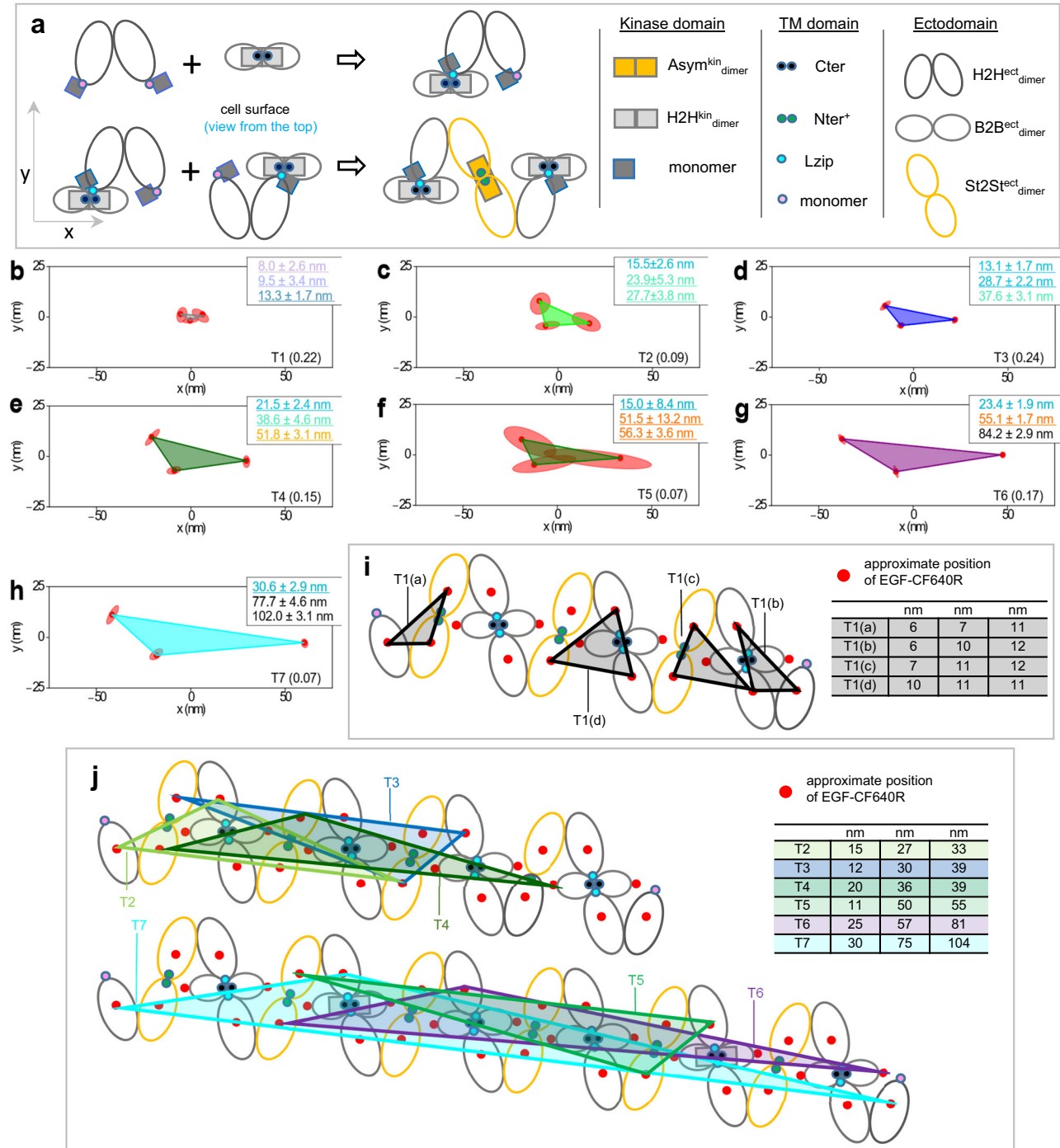

**Fig. 7 | The extracellular structure of ligand-free hetero^conf-oligomers. a** Top left, Tetramer assembled by an H2H^ect/2x^kin_monomers sub-unit and a B2B^ect/H2H^kin_dimer sub-unit. A transmembrane monomer of an H2H^ect/2x^kin_monomers interacts with the transmembrane dimer of the B2B^ect/H2H^kin_dimer via an Lzip interface. The Lzip interface is related to the transmembrane dimers in the B2B^ect/H2H^kin_dimer sub-unit and St2St^ect/Asym^kin_dimer sub-unit by a rotation of the helix along the long axis[45]. Bottom left, two tetramers form a St2St^ect/Asym^kin_dimer sub-unit. Note that the latter could in principle form also from one H2H^ect/2x^kin_monomers sub-unit forming an Lzip interface with a tetramer made of one H2H^ect/2x^kin_monomers sub-unit and a B2B^ect/H2H^kin_dimer sub-unit. (Attempts to join a H2H^ect_dimer/2x^kin_monomers sub-unit and a St2St^ect/Asym^kin_dimer sub-unit via Lzip contacts led to steric clashes. More details in Supplementary Fig. 4). Right, Annotated interfaces in the model. **b–h** Seven distinct optimally grouped triangles from 2D FLImP separations determined between EGF-CF640R probes bound to cell surface T766M-EGFR (Supplementary Fig. 5). The side lengths of each triangle are annotated in each inset in the colors used in 1D FLImP decompositions when found in the separation sets of either T766M-EGFR or WT-EGFR (Fig. 4a, d), and colored and underlined if found in both. Separations >70 nm are outside the range of 1D FLImP. The abundance of each triangle (T1-7) is stated bottom right. **i** Ligand-free hetero^conf-oligomer model extended as described in **a**. The approximate positions where EGF-CF640R would bind DIII of the ectodomains are marked (red circles). Four versions of (b), the experimentally-optimized triangle 1 within errors (T1a-T1d), are superimposed. Inset, table of the side lengths of the superimposed triangles. **j** Top, ligand-free hetero^conf-oligomer model with the triangle groups T2-T4 (**c–e**) superimposed. Bottom, hetero^conf-oligomer model with the largest triangles T5-T7 (**f–h**) superimposed. Inset, table of the side lengths of the superimposed triangles.

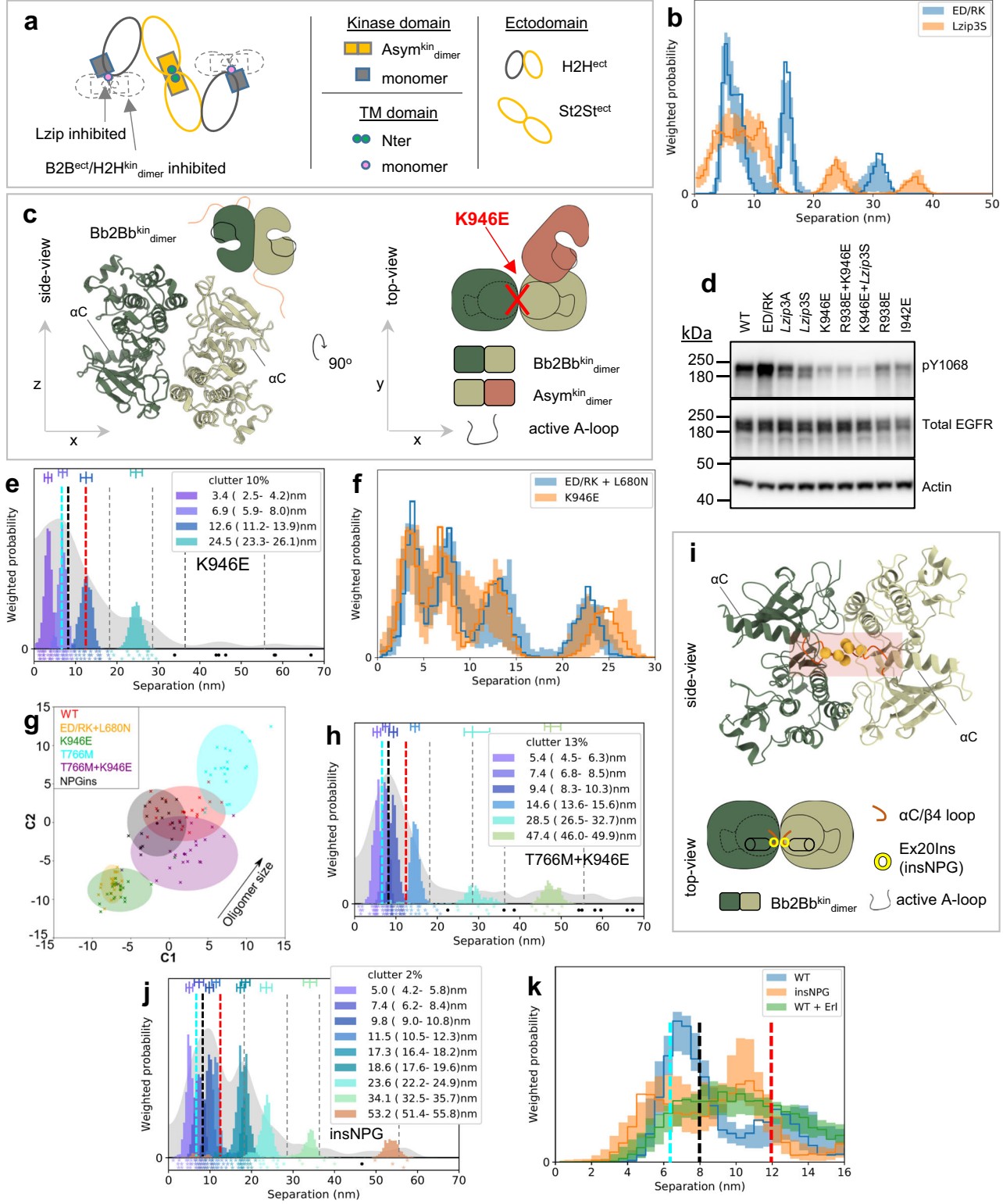

hetero[conf]-oligomer structure but not phosphorylation we focused on the T766M + H566F and omitted the apparently equivalent T766M +Lzip3S (Supplementary Fig. 6). From the mutations that disrupt oligomer structure and phosphorylation we chose T766M + K946E over T766M + I942E because unlike the I942E mutation, K946E does not interfere with the ligand-bound state (Supplementary Fig. 10b).

Next, we compared the ability of these Ba/F3 cell lines to establish and grow tumors. Notably, all these cell lines also stably expressed GFP, exploited to image growing tumors. We found a lag-time prior to

the onset of palpable tumors of around 20 days in all cohorts, after which the tumors formed in animals that had received Ba/F3 + T766M and Ba/F3 + T766M + H566F cells and, with an additional delay, in the Ba/F3 + T766M + K946E cohort (Fig. 10a). Importantly, mirroring the growth pattern observed in vitro, Ba/F3 + T766M tumors grew best throughout, followed by the double mutant tumors (Ba/F3 + T766M + H566F > Ba/F3 + T766M + K946E), while WT-EGFR tumors did not establish or grow. The lack of tumor cell survival in the Ba/F3 + WT cohort was already strongly suggested by day 21 when we observed no

**Fig. 8 | A Bb2Bb$^{kin}_{dimer}$ sub-unit underpins activation in ligand-free oligomers. a** Tetramer that would form if B2B$^{ect}$/H2H$^{kin}_{dimer}$ sub-units or Lzip contacts were inhibited (interfaces annotated). **b, f, k** Comparisons between decomposed separation probability distributions between FLImP datasets. Continuous lines show the marginalized separation posterior sum of the abundance-weighted peaks, for each condition in the inset. Fluctuations around each continuous line arise from variations from FLImP decompositions for 20 bootstrap-resampled datasets to assess errors from finite measurements number. **c** Structure and cartoon of the Bb2Bb$^{kin}_{dimer}$ formed mainly through N-to-C lobe interactions (PDB ID 3VJO [https://www.rcsb.org/structure/3VJO])[47]. A kinase monomer can dock in the Bb2Bb$^{kin}_{dimer}$ to form an Asym$^{kin}_{dimer}$, as shown, with the position of the A-loop marked. The K946E mutation breaks the Bb2Bb$^{kin}_{interface}$ interface (Supplementary Fig. 7a, Supplementary Note 1). **d** Western blot showing phosphorylation in the absence of ligand in transfected CHO cells. (Blot quantification in Supplementary Fig. 7b, representative of n = 3 blots). **e, h, j** FLImP analysis of 100 separation probability distributions between Affibody-CF640R pairs. The sum of posteriors of individual separations (gray background) and abundance-weighted probability distributions of individual components of decomposed separation distribution (colored peaks). Plot legend and bars above colored component distributions give the median and most-compact 68% confidence interval for each. Stars and dots below show individual separations assigned to peaks (colored) or clutter (black). Legend also gives median proportion of measurements assigned to clutter. Dashed lines mark WT-EGFR median peak positions (also in (k)). **g** Wasserstein MDS analysis of FLImP decompositions. Similarities or dissimilarities between the 21 separation sets of different conditions (one main FLImP decomposition plus 20 bootstrap-resampled decompositions) are compared. The plot axes are components C1 and C2. C1 represents the dimension that captures the largest amount of data variance, while C2 represents the second-largest amount of variance orthogonal to C1. The ellipse centers (95% confidence range) mark the positions of the main FLImP decompositions. The crosses mark those of individual bootstrap-resampled separation sets. **i** The Bb2Bb$^{kin}_{dimer}$ sub-unit[47] (side and top-view). Inserted residues of the Ex20Ins mutation on the αC/β4 loop shown (yellow circles). Source data provided as a 'Source data' file.

---

GFP reporter signals at the injection site (Supplementary Fig. 10c–e). The experimental endpoint was on day 41 when we imaged all animals by IVIS (Fig. 10b, c) to determine tumor fluorescence as an independent measure of tumor growth between cohorts. We found significant differences in fluorescence signals between groups, in line with caliper measurements (Fig. 10a), except for Ba/F3 + T766M vs. Ba/F3 + T766M + H566F tumors. The latter highlights the limitations of epifluorescence imaging, which did not reflect accurately the signal differences between larger tumors due to limited tissue penetration and absorption within thicker tissues. Harvested tumors were first qualitatively imaged under daylight (Fig. 10d) and then weighed (Fig. 10e), upon which the significant growth differences between the Ba/F3 + T766M vs. Ba/F3 + T766M + H566F tumors were evident, as were the other differences already seen in vivo in Fig. 10a. Moreover, we subjected harvested tumor tissues to histology to demonstrate tumor morphology by H&E (Fig. 10f) and pan-EGFR expression by anti-EGFR staining (Fig. 10g vs. staining control in Supplementary Fig. 10f). All tumors were positive for EGFR with plasma membrane localization clearly visible in stained tissues (Fig. 10g). Receptor copy numbers of ~2×10$^5$ per cell were estimated by comparison with tumor xenografts from other cell lines (Supplementary Fig. 10g). Our results are consistent with previous intravital fluorescence imaging data that showed predominant localization of EGFR at the plasma membrane consistent with very low ligand binding[49].

Notably, the T766M + H566F mutant, which does not reduce basal phosphorylation compared with T766M (Supplementary Fig. 7c), also handicaps tumor growth. An explanation is that T766M + H566F expressing Ba/F3 cells are unable to phosphorylate and activate AKT signaling to the same extent as T766M (Supplementary Fig. 7c).

## Discussion

We combined super-resolution FLImP imaging with in silico modeling and mutagenesis to identify the interfaces that assemble ligand-free, kinase active EGFR oligomers. The <3 nm resolution achieved allowed us not only to fingerprint previously proposed ligand-free dimer conformers but, crucially, show how interactions between kinase inactive dimers assemble ligand-free hetero$^{conf}$-oligomers that bear active St2St$^{ect}$/Asym$^{kin}_{dimer}$ sub-units. From this knowledge and the known shapes of the ligand-free dimer conformers we derived a structural model of ligand-free hetero$^{conf}$-oligomerization that explains how EGFR can achieve ligand-independent auto-phosphorylation. In addition, the extracellular part of this model was validated by implementing a 2D FLImP version.

The tethered conformation of EGFR's ectodomain appeared during the evolution of vertebrates[68] and was suggested to have evolved to prevent crosstalk between the different EGFR homologs in the vertebrate EGFR family[69]. Our MD simulations and FLImP results propose another biological role for the tethered conformation, which is to regulate the formation of the H2H$^{ect}_{dimer}$, and through it, hetero$^{conf}$-oligomer size, receptor activation, and, in the pathological context of T766M-EGFR, tumor formation.

Comparing models and data revealed that a previously orphan Bb2Bb$^{kin}_{interface}$ reported by X-ray crystallography[47] plays a regulatory role in ligand-free hetero$^{conf}$-oligomerization and activation. The role of the Bb2Bb$^{kin}_{interface}$ is to buttress the Asym$^{kin}_{dimer}$, leading to activation once the B2B$^{ect}$/H2H$^{kin}_{dimer}$ cantilevers H2H$^{ect}_{dimer}$/2x$^{kin}$ monomers into position to form the St2St$^{ect}$/Asym$^{kin}_{dimer}$. This mechanism explains how the Asym$^{kin}_{dimer}$ can form in the absence of ligand-induced conformational changes that are typically required to overcome the activation barrier associated with the formation of the asymmetric interface between activator and receiver kinases[42]. Importantly, our work shows that Ex20Ins mutations stabilize the Bb2Bb$^{kin}_{interface}$, providing a breakthrough in our understanding of these mutations, which so far have been studied only at the monomeric level.

Comparing models and data also suggested a second S2S$^{kin}_{interface}$ that we identified from a crystal contact in a structure of the H2H$^{kin}_{dimer}$[54]. This interface allows the H2H$^{kin}_{dimer}$ to hijack kinase monomers, preventing formation of the stimulatory Bb2Bb$^{kin}_{interface}$, and down-regulating activation. The S2S$^{kin}_{interface}$ stabilizes the H2H$^{kin}_{dimer}$, explaining its autoinhibitory role. We propose that the regulation of the activation of ligand-free hetero$^{conf}$-oligomers rests on balancing the interactions between the Asym$^{kin}_{dimer}$ and the H2H$^{kin}_{dimer}$ with the Bb2Bb$^{kin}_{interface}$ and the S2S$^{kin}_{interface}$.

Beyond its autoinhibitory role, the H2H$^{kin}_{dimer}$ also acts as a scaffold promoting the formation of larger hetero$^{conf}$-oligomers. Thus, by stabilizing the H2H$^{kin}_{dimer}$, the T766M mutation decouples oligomer growth from its dependence on the autoinhibitory S2S$^{kin}_{interface}$. This indirectly underpins the formation of Asym$^{kin}_{dimer}$ units without the need to hijack monomers from the Bb2Bb$^{kin}_{interface}$, which can thereby still buttress Asym$^{kin}_{dimer}$ sub-units within larger oligomers, thus amplifying ligand-independent phosphorylation for the T766M receptor mutant.

As an example, we used T766M-EGFR-dependent cell growth to evaluate in vivo the effect of disrupting ligand-free hetero$^{conf}$-oligomer structure by inhibiting the tethered conformation and the Bb2Bb$^{kin}_{interface}$. Excitingly, disrupting the Bb2Bb$^{kin}_{interface}$, which has a deleterious effect on oligomer size and phosphorylation, almost abolishes tumor growth. A clue to a possible mechanism is provided by previous in vivo intravital microscopy studies that proposed that the small pool of high-affinity EGFRs drives tumor xenograft growth when stimulated by the 17–100 pM ligand concentration surrounding the tumor cells[49]. We found that St2St$^{ect}$/Asym$^{kin}_{dimer}$ sub-units are high-affinity ligand binding sites. According to our model, oligomers display ~25–30% high-affinity St2St$^{ect}$/Asym$^{kin}_{dimer}$ sub-units. In CHO cells

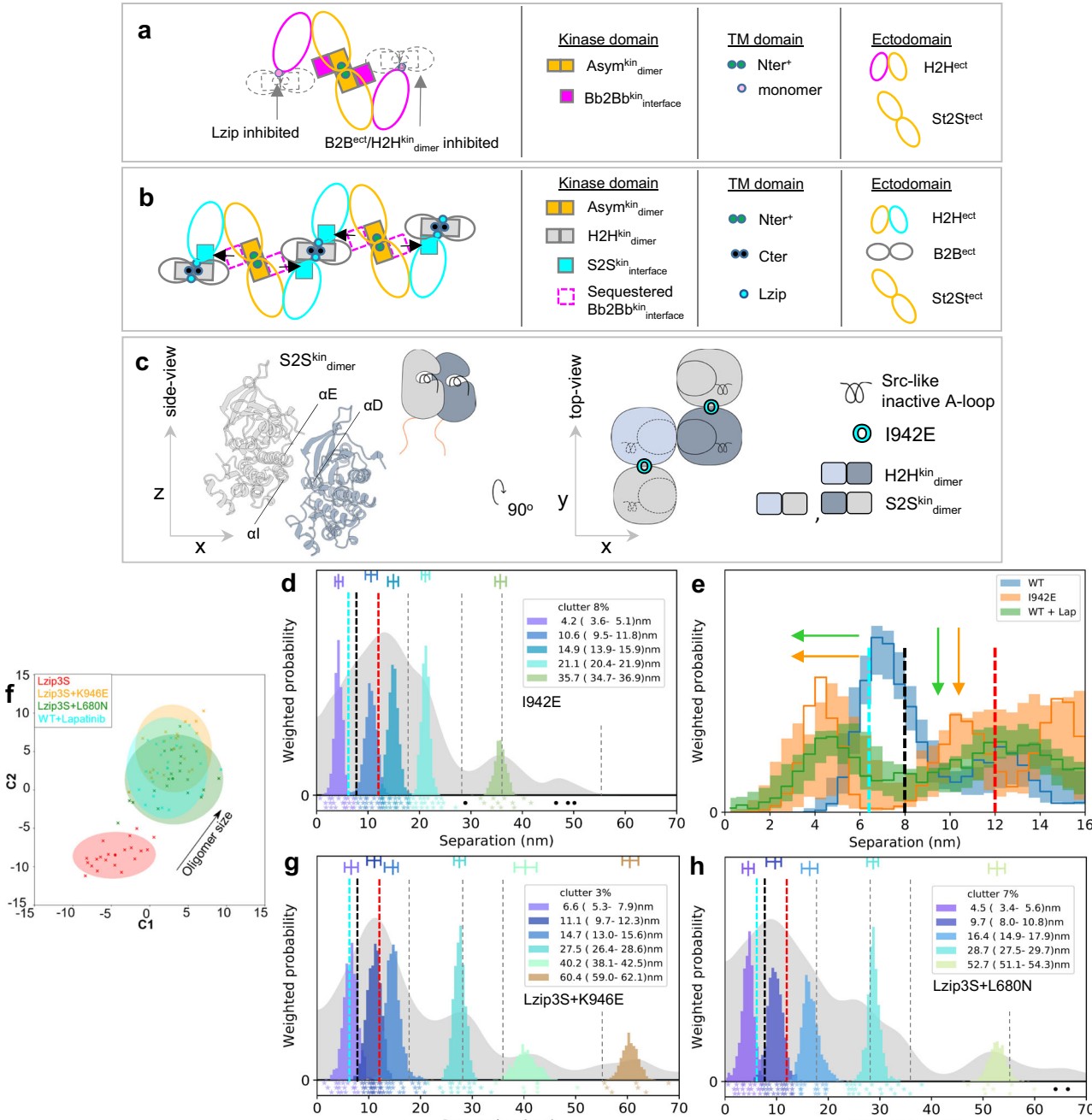

**Fig. 9 | An autoinhibitory S2S$^{kin}_{dimer}$ sub-unit promotes hetero$^{conf}$-oligomer growth. a** Tetramer that would form after inhibiting B2B$^{ect}$/H2H$^{kin}_{dimer}$ sub-units or Lzip contacts with two Bb2Bb$^{kin}_{interfaces}$ buttressing an Asym$^{kin}_{dimer}$ sub-unit. Ecto-domains and kinases belonging to the same receptor in H2H$^{ect}_{dimer}$/2x$^{kin}_{monomers}$ sub-units are colored accordingly. **b** Ligand-free hetero$^{conf}$-oligomer model incor-porating four S2S$^{kin}_{interfaces}$. **c** Left, side-view of the S2S$^{kin}_{dimer}$ sub-unit maintained mainly through the β2-sheet and αD-helix of the one monomer and the αE- and αI-helices of another (PDB ID: 5CNO [https://www.rcsb.org/structure/5CNO])[48]. The position of the A-loop in the inactive configuration is marked. Right, cartoon representation of the top-view of an H2H$^{kin}_{dimer}$ sub-unit flanked by two monomers via S2S$^{kin}_{interfaces}$. **d, g, h** FLImP analysis of 100 separation probability distributions between Affibody-CF640R pairs in the conditions indicated: Sum of posteriors of individual separations between fluorophores (gray background) and abundance-weighted probability distributions of individual components of decomposed separation distribution (colored peaks). Plot legend and bars above colored com-ponent distributions give the median and most-compact 68% confidence interval for each. Legend also gives median proportion of measurements assigned to

clutter. The median peak positions marked by dashed lines are those of WT-EGFR. **e** Comparisons between decomposed separation probability distributions between datasets. The continuous lines show the marginalized separation posterior, i.e. the sum of the abundance-weighted peaks, for each condition in the inset. The fluc-tuations around each continuous line arise from variations derived from FLImP decompositions for 20 bootstrap-resampled datasets to assess errors due to finite number of measurements. Dashed lines marked WT-EGFR median peak positions (as in **d**). Colored arrows show shift from WT-EGFR. **f** Wasserstein MDS analysis of FLImP decompositions for the conditions in the inset. Similarities or dissimilarities between the 21 separation sets of different conditions (one main FLImP decom-position plus 20 bootstrap-resampled decompositions) are compared. The plot axes are components C1 and C2. C1 represents the dimension that captures the largest amount of variance in the data, while C2 represents the second-largest amount of variance that is orthogonal to C1. The ellipse centers (95% confidence range) mark the positions of the main FLImP decompositions. The crosses mark the positions of individual bootstrap-resampled separation sets.

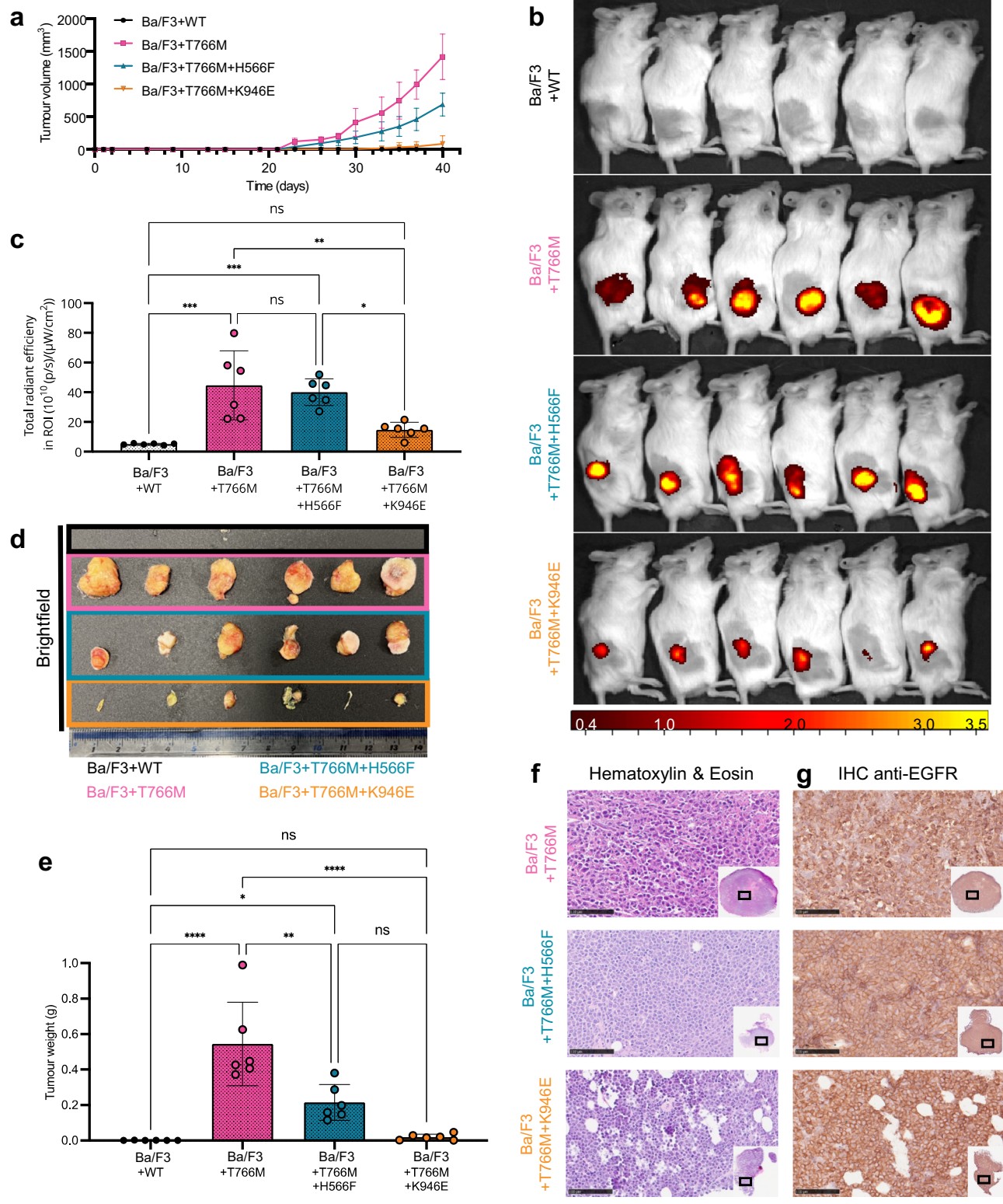

expressing ~10[5] EGFR copies/cell we estimated that ~15-40% of cell surface EGFRs are incorporated in ligand-free oligomers. Similar oligomer fractions would be expected for Ba/F3 + T766M tumor xenograft cells, which we found to express ~2 × 10[5] EGFR copies/cell[70]. Among the oligomer species, ~1/2 are hexamers and larger oligomers[36] that can bear at least one St2St$^{ect}$/Asym$^{kin}_{dimer}$ sub-unit. Thus, our model predicts that ~2–6% of the total sites will be high-affinity St2St$^{ect}$/Asym$^{kin}_{dimer}$ sub-units, consistent with previous predictions[71–73]. T766M-EGFR oligomers are larger and more stable[36], displaying more

high-affinity St2St$^{ect}$/Asym$^{kin}_{dimer}$ sites. The finding that inhibition of the Bb2Bb$^{kin}_{interface}$, which abolishes the assembly of high-affinity St2St$^{ect}$/Asym$^{kin}_{dimer}$ sub-units, also abolishes tumor growth is therefore consistent with the premise that high-affinity sites drive tumor growth when stimulated by ligand. We note that the decreased accessibility of the ligand to its binding site in H566F-EGFRs might also explain the partial effect of this mutation in reducing tumor growth.

By revealing how tumor-driving high-affinity St2St$^{ect}$/Asym$^{kin}_{dimer}$ sub-units assemble within oligomers, our work has also revealed an

**Fig. 10 | Tumors established from Ba/F3 cell lines with wild-type or indicated mutant EGFR showed different growth behavior in vivo. a** Tumor growth measurements by calipers: immunodeficient (NSG) mice were used to establish tumors in their flanks and tumor establishment and cumulative growth was monitored using calipers. $N = 6$ biologically independent animals per cohort, mean values plotted with error bars representing standard deviation (SD). **b** IVIS imaging of all animals from (a) at the experimental endpoint (day 41) showing epifluorescence signals from animal tumors. Radial efficiency was calculated as $10^{10}$(p/s/cm²/sr)/(μW/cm²). **c** GFP fluorescence signal quantification of tumors corresponding to $N = 6$ biologically independent animals in **b**. Statistical analysis by 1-way ANOVA ($\alpha = 0.05$ and Tukey's multiple comparison correction). * = 0.012; ** =

0.0029; *** = 0.0001-0.0006 (Supplementary Table 5). Data are presented as mean values; error bars are SD. **d** Harvested tumors from animals with established tumors were photographed under daylight. No tumors had established in the cohort that had received Ba/F3 + WT cells. **e** Tumor weights from harvested tumors of $N = 6$ biologically independent animals. Statistical analysis by 1-way ANOVA with Tukey's multiple comparison correction: * = 0.0416; ** = 0.0013; **** <0.0001 (Supplementary Table 5; data are presented as mean values; error bars are SD). **f** Hematoxylin and eosin staining of tumors; scale bars are 100 μm. **g** Immunohistochemistry staining using a pan anti-EGFR antibody; corresponding background control staining is shown in Supplementary Fig. 10f. Scale bars are 100 μm. Source data are provided as a 'Source data' file.

Achilles heel in drug-resistant NSCLC tumors that could be therapeutically targeted, counterintuitively, with drugs or small proteins that interfere with the tether, and/or can inhibit the Bb2Bb$^{kin}_{interface}$. Furthermore, these interfaces are away from the ATP-binding pocket mutational hotspot. Targeting protein-protein interactions is an alternative direction in treating diseases and an essential strategy for drug development[74]. Given the poor long-term efficacy of current treatments for NSCLC, the structural understanding from this work of how T766M-EGFR and Ex20Ins-EGFR amplify cell growth suggests a possible route for more effective therapies. Notably, mutant, and WT-EGFR share hetero$^{conf}$-oligomer structure. It would be interesting to find out whether these principles apply to cancers driven by EGFR overexpression which currently have very limited therapeutic options.

## Methods

### Reagents
Antibodies and reagents were purchased as follows: mAb-2E9 (Abcam ab8465, RRID: AB_2096462), Anti-EGFR Affibody® Molecule (Abcam ab95116; RRID: AB_11156238), Anti-EGFR (D38B1, Cell Signaling Technology (CST) 4267; RRID: AB_2246311), Anti-Phospho-Akt (Ser473, D9E, CST 4060; RRID: AB_2315049), Anti-beta-Actin Monoclonal Antibody, HRP Conjugated (13E5 CST 5125; RRID: AB_1903890), Anti-Phospho-EGF Receptor (Tyr1068, D7A5 CST 3777; RRID: AB_2096270), Anti-EGFR (R & D Systems, AF231; RRID: AB_355220), Anti-EGFR (phospho Y992, EM-12, Abcam ab81440; RRID: AB_1658463), Anti-mouse IgG-HRP (Jackson ImmunoResearch 715-035-150; RRID: AB_2340770), Anti-rabbit IgG-HRP (Jackson ImmunoResearch 711-035-152; RRID: AB_10015282), Anti goat IgG-HRP (Jackson ImmunoResearch 705-035-147; RRID: AB_2313587), Anti-Rabbit-HRP (Dako P044801, RRID: AB_2617138), Recombinant Murine EGF (Peprotech, 315-09), Recombinant Murine IL-3 (Peprotech, 213-13), Erlotinib Hydrochloride (Biovision, 1558-100), Lapatinib Ditosylate (Biovision, 1642-25), 4% Paraformaldehyde (EM Grade, Electron Microscopy Service (EMS), 157-4), 25% Glutaraldehyde (Grade I, Sigma G5882), FluoSpheres™ (carboxylate modified, 0.1 μm, infrared (715/755), Invitrogen F8799).

### Cell culture
mAb-108 expressing hybridoma cells (ATCC, HB-9764) were grown in high glucose DMEM media, no phenol red (Gibco, 31053044) supplemented with 10% fetal bovine serum (FBS – Gibco, 10270-106) and 1 mM sodium pyruvate (Gibco, 11360070). Chinese Hamster Ovary (CHO) cells (gift from Prof. Peter Parker at The Francis Crick Institute, UK) were grown in DMEM/F12 with no phenol red (Gibco, 21041-025) + 10% (v/v) FBS + 1% penicillin/streptomycin (Gibco, 15140148) and when made to stably express EGFR or mutants the media was supplemented with 4 μg/mL puromycin (Gibco, A1113803) to maintain expression. CHO cells expressing ΔC-EGFR (gift from Prof. Linda Pike, Washington University School of Medicine, USA) were grown in phenol-red-free DMEM supplemented with 10% FBS, 2 mM glutamine (Gibco, 25030081), 1% penicillin/streptomycin, 100 μg/mL hygromycin (Gibco, 10687010) and 100 μg/mL geneticin (Gibco, 10131035). Ba/F3 cells (Creative Biogene, CSC-C2045) stably expressing EGFR or EGFR

mutants were grown in RPMI1640, no phenol red (Gibco, 11835063), with 2 mM L-glutamine and without HEPES + 10% heat-inactivated FBS (Gibco, 10500064) + 10 ng/mL mouse IL3 (Peprotech, 213-13) + 1% penicillin/streptomycin + 2 μg/mL puromycin. All cell lines were verified negative for mycoplasma before use and tested routinely.

### Plasmid construction & mutagenesis
Point mutations in EGFR plasmids were introduced using Quikchange Lightning Site-directed mutagenesis kit (Agilent Technologies – cat. no. 210518-5) using primer pairs listed in Supplementary Table 4. All constructs were verified by sequencing the whole coding sequence of EGFR. To generate EGFR PiggyBac plasmids, WT-EGFR was PCR amplified from WT-EGFR/pcDNA3 plasmid using primer pairs with NheI and NotI restriction enzyme sites (Supplementary Table 4) and inserted into PB513B-1 vector using standard molecular biology techniques.

### Purification of mAb-108 from cell culture media
mAb-108 antibody was purified from mAb-108 hybridoma cell culture supernatant using a mouse TCS antibody purification kit (Abcam, ab128749). Purified antibody was quantified using a Nanodrop and stored at 4 °C.

### Generation of EGFR CHO stable cell lines
CHO cells expressing one of the following were generated: WT-EGFR, ED/RK-EGFR, T766M-EGFR, T766M + K946E-EGFR or T766M + I942E-EGFR. CHO cells were transfected with a mix of 0.1 μg of Super Piggybac Transposase expression vector (PB210PA-1, System Biosciences) and 1 μg of the appropriate endotoxin-free EGFR plasmid DNA (in PB513B-1 vector, System Biosciences) using FuGENE HD at 1:3 DNA:FuGENE HD ratio according to manufacturer's instructions. The cells were selected in fresh media containing 4 μg/mL puromycin for 7-10 days. Surviving clones of cells were pooled and checked for EGFR expression by western blotting and confocal imaging.

### Generation of Ba/F3 stable cell lines
Ba/F3 cell lines were electroporated using a Neon transfection kit (Invitrogen, MPK10096) and MicroPorator device (Invitrogen). On the day of electroporation, cells were washed once in PBS and resuspended in buffer R at a final concentration of $1.5 \times 10^7$ cells/mL. 100 μL cells ($= 1.5 \times 10^6$ cells) were electroporated with a mix of 0.5 μg of Super Piggybac Transposase expression vector and 5 μg of the appropriate endotoxin-free EGFR plasmid DNA according to the manufacturer's instructions. Cells were selected in media containing 2 μg/mL Puromycin at a density between 0.3 to $1 \times 10^6$ cells/mL for 7–10 days to obtain polyclonal cell populations stably expressing EGFR.

### Ba/F3 IL3-independent growth assay
Ba/F3 cells stably expressing EGFR or EGFR mutants were washed once in PBS and resuspended in Ba/F3 media without IL3 and puromycin. Cells were grown in the absence of IL3 for 5 days, then seeded at a density of 20,000 cells (in 100 μL of media) per well in white-bottomed

96-well plates in triplicate. As a positive control for cell growth, additional cultures of each cell line were maintained in complete media (including 10 ng/mL recombinant murine IL3). These cells were seeded as above in the presence of IL3. Cell viability was measured every day for 4 days for the positive controls, or 10 days for the cells without IL3, using CellTiter-Glo Luminescent Assay (Promega, G7572) and a CLARIOstar plus microplate reader according to manufacturer's instructions.

## FLImP sample Affibody labeling

CHO cells stably expressing WT-EGFR or mutant EGFR were seeded in 3 of the central 4 wells of μ-Slide 8 well high glass bottom slides (ibidi, 80807), coated with 1% BSA, at $1.8 \times 10^4$ cells per well and allowed to grow for 2 days. The 4th well (top left) was coated with poly-L-Lysine (PLL – Sigma P4707-50ml) only. ΔC-EGFR-expressing CHO cells were grown in the presence of 50 ng/ml doxycycline (ThermoFisher Chemicals, J67043.AD). Cells were cultured for 48 h prior to labeling.

Transient transfections of WT CHO cells using FuGENE HD (Promega, E2312) and 600 ng plasmid DNA, at 1:3 DNA:FuGENE ratio, were necessary for the following EGFR mutants: H566F, Lzip3S, ED/RK + L680N, ED/RK + L680N + Lzip3S, K946E, insNPG, G564P, G564P + ED/RK, Lzip3A, T766M + Lzip3S and T766M + I942E. Transfections were carried out 24 h after seeding and left a further 24 h prior to labeling.

Samples were starved in low serum medium (0.1% FBS), with 1 μM Lapatinib or Erlotinib if necessary, for 2 h before rinsing with PBS (PBS without $Ca^{2+}$ and $Mg^{2+}$ was used throughout). Samples were then chilled on ice at 4 °C for 10 min in PBS and labeled with 8 nM HER1 Affibody-CF640R, with 1 μM Lapatinib or Erlotinib if necessary, for 1 h on ice at 4 °C. Cells were rinsed and fixed with 3% paraformaldehyde plus 0.5% glutaraldehyde for 15 min on ice and 15 min at room temperature (RT).

Cells were rinsed with PBS and stained with 1 μg/mL Hoechst (Invitrogen, H21492) in PBS at RT for 10 min then rinsed again with PBS. Samples were stored at 4 °C and prior to imaging slides were brought up to RT. PLL was removed from the top left well and PBS from the samples wells. In all wells, 300 μLs of FluoSpheres™, 0.1 μm, infrared (715/755) (Invitrogen F8799) were added (1/50,000 dilution of stock in PBS) to be used as fiducials.

Samples were loaded onto the ONI nanoimager microscope so that the central dividing cross between the central 4 wells of the slide corresponded to position 0,0. The slide was warmed to 34 °C prior to sample collection and the sample was brought into initial focus by the operator.

## FLImP sample EGF labeling

Samples were labeled as above except for the following: for mAb-108 and mAb-2E9 treatment, CHO cells expressing WT-EGFR were starved of serum for 2 h then treated with 200 nM mAb on ice at 4 °C for 2 h. For 2D FLImP triangles, CHO cells expressing T766M-EGFR were serum starved for 2 h. Cells were fixed in 3% paraformaldehyde in PBS for 15 min at RT and rinsed. WT-EGFR cells and T766M-EGFR cells were labeled with 10 or 20 nM EGF-CF640R respectively for 1 h at RT and rinsed with PBS. The samples were fixed again for 15 min at RT with 3% paraformaldehyde plus 0.5% glutaraldehyde.

## FLImP data acquisition

Image acquisition was performed using an Nanoimager S (ONI Oxford, UK) single molecule imaging microscope with a 1.49 N oil immersion objective, operating NanoImager software (Version: 1.7.3.10248−ef4ff2c0). Single frame FLImP acquisitions used an 8 mW 640 nm diode laser for 20 ms exposure, single frame Hoechst acquisitions used an 8 mW 405 nm diode laser with 20 ms exposure.

The TIRF angle was set, and the instrument temperature was maintained at 34 °C. Data acquisition was facilitated using the PythONI (Python API from ONI) included with the NanoImager software suite. Automated data acquisition procedures are described in Supplementary Note 2. Fiducials in the top left well of each plate were used to estimate PSF properties from a $3 \times 3$ frame region of interest (ROI). The focal plane was established using the provided ONI autofocusing software (followed by a focal polishing step as described in Supplementary Note 2).

## Western blot

CHO cells, 24 h post-transfection with FuGENE HD as described above, were serum starved in media containing 0.1% FBS for 2 h at 37 °C. Cells were placed on ice, washed in cold PBS and lysed directly in 6-well dishes in cell lysis buffer [50 mM Tris/HCl (pH 7.4), 1 mM EDTA, 1 mM EGTA, 50 mM sodium fluoride, 5 mM sodium pyrophosphate, 10 mM sodium β-glycerol 1-phosphate, 1 mM dithiothreitol, 1 mM sodium orthovanadate, 0.27 M sucrose, 1% (v/v) Triton X-100, 1x Protease inhibitor]. Cell extracts were clarified by centrifugation, and the protein concentration was determined using Bradford assay. 10–20 μg of total cell lysate was resolved on an 8% Bolt Bris-Tris gel, and proteins were transferred to PVDF membrane. Membranes were blocked and probed with primary and secondary-HRP antibodies according to manufacturer's instructions. Subsequently, membranes were incubated with Immobilon ECL Ultra Western HRP substrate solution (Millipore WBKLS0500) and imaged on Biorad Chemidoc MP Imager. Images were quantified in ImageLab software (Biorad) where intensity of each band was relatively quantified against WT-EGFR which was set to 1.

## Single particle tracking

Cells were seeded on 1% BSA-coated 35 mm no. 1.5 (high precision) glass-bottomed dishes (MatTek) in 2 mL of media. After 24 h, transient transfections were performed using FuGENE HD as described above, and cells were grown for further 24 h. Prior to imaging, cells were starved for 2 h at 37 °C in 0.1% FBS. Cells were then rinsed twice with 0.1% FBS pre-heated at 37 °C and were labeled with a 1:1 mixture of 8 nM Affibody- Alexa 488 / Affibody- CF640R for 7 min at 37 °C. Cells were rinsed twice with low serum medium pre-heated at 37 °C and promptly imaged as described previously[36,75]. Typically, for each condition, at least 30 field of views comprising one or more cells were acquired from a total of at least 3 independent biological replicates.

All single-molecule time series data (for FLImP and single particle tracking) were initially analyzed using the multidimensional analysis software described previously[76]. The colocalization event duration analysis was performed as before[36].

## Confocal imaging

For all confocal experiments, cells expressing WT-EGFR, L680N-EGFR or ED/RK-EGFR were seeded, serum-starved for 2 h as described above and rinsed twice in ice-cold PBS and cooled down for 10 min on ice.

For anti-EGFR Affibody and EGF competition binding experiments, cells were then pre-treated with either 200 nM, 400 nM or 600 nM of ice-cold Affibody-CF640R in PBS or with mock treatment (PBS) for 1 h at 4 °C, rinsed with ice-cold PBS and fixed with 3% paraformaldehyde in PBS for 30 min at 4 °C. After fixation, cells were rinsed again with RT PBS and labeled with 400 nM EGF-Alexa488 in PBS for 1 h at RT, rinsed with RT PBS, then fixed with 3% paraformaldehyde + 0.5% glutaraldehyde for 15 min at RT.

For the mAb-2E9 (Abcam, ab8465, RRID: AB_2096462) binding experiments, after starvation cells were pre-treated with 200 nM of mAb-2E9-AF488 in PBS or with mock treatment (PBS) for 2 h at 4 °C, rinsed with ice-cold PBS and fixed with 3% paraformaldehyde in PBS for 15 min at 4 °C. For the EGF binding post-fixation test, cells were labeled with 200 nM EGF-CF640R for 2 h at 4 °C then fixed.

After fixation, cells were rinsed again with RT PBS and labeled with 200 nM EGF-CF640R in PBS for 1 h at RT, then rinsed with PBS, fixed with 3% paraformaldehyde + 0.5% glutaraldehyde for 15 min at RT and rinsed with PBS. All samples were stored in PBS at 4 °C until the time of acquisition, and allowed to pre-warm at RT, before loading on the microscope.

Image acquisition was performed on an Elyra PS1, using Zen Black v2.3 SP1 using 633 nm or 488 nm laser excitation.

Colocalization analyses were carried out on images of 600 μm optical slices and performed using Huygens software (Scientific Volume Imaging).

For the mAb-2E9 binding experiments, pixel-wise intensity or intensity ratio distributions were extracted from the data using Huygens (SVI). The non-parametric Kruskal-Wallis statistical test was performed and T-test post-hoc analysis with Bonferroni multiple comparison correction was applied to calculate P Values in Python.

## Mice tumor models

In this study, 24 young adult male (6-7 weeks old, $24.6 \pm 2.1$ g) were used for all animal experiments (NOD.Cg-Prkdc$^{scid}$ Il2rg$^{tm1Wjl}$/SzJ mice mice, Charles River UK, Strain code: 614; RRID:IMSR_JAX:005557). All mice were maintained within the King's College London Biological Services Unit under specific pathogen-free conditions in a dedicated and licensed air-conditioned animal room (at $23 \pm 2\,°C$ and 40-60% relative humidity) under light/dark cycles lasting 12 h every day. They were kept in individually ventilated standard plastic cages (501cm$^2$ floor space; from Tecniplast) including environmental enrichment and bedding material in the form of sterilized wood chips, paper stripes and one cardboard roll per cage. Maximum cage occupancy was five animals, and animals were moved to fresh cages with fresh environmental enrichment and bedding material twice per week. Sterilized tap water and food were available *ad libitum*; food was PicoLab Rodent Diet 20 (LabDiet) in the form of $2.5 \times 1.6 \times 1.0$ cm oval pellets that were supplied at the top of the cages.

Male NSG mice were used to establish subcutaneous tumor models (in right flanks) with indicated stable Ba/F3 cell lines. Males were used because lung cancer has a higher incidence in human males. After acclimatization, mice were randomly allocated into four cohorts with six individuals each, shaved on their flanks, and then each received $2 \times 10^6$ tumor cells suspended in 100 μL PBS subcutaneously. Tumor growth was followed by calipers and tumor volumes calculated. Tumor models were grown to compare tumor growth between cohorts. The experimental endpoint was defined by the time the humane endpoint was reached for the cohort with the largest tumor growth, and then all animals were sacrificed. All experimental protocols were monitored and approved by the King's College London Animal Welfare and Ethical Review Body in accordance with UK Home Office regulations (Project License PP4067431) under the Animals (Scientific Procedures) Act 1986 and UK National Cancer Research Institute (NCRI) Guidelines for the Welfare and Use of Animals in Cancer Research. The mice were treated according to the endpoints stated in the Project License. The mice were well throughout, and the tumor volumes were established using the formula for an ellipsoid, $V = ½LxW^2$. The volumetric size limit is 1.5 cm$^3$ which determined the end point of the experiment.

## In vivo imaging of tumor models

In vivo GFP fluorescence imaging of superficial tumor models was performed to visualize tumor growth in some animals per group over time and to quantify tumor growth differences in all animals at the experimental endpoint. ROIs were manually drawn including the whole tumor (or the injection sites where no tumors were visible) and used to calculate the radiant efficiency. Prism software version 9 (GraphPad, La Jolla, USA) was used to calculate all statistical parameters as indicated. Generally, p-values were calculated using significance levels of $\alpha = 0.05$. In-text numbers indicate means of pooled data ± standard deviation (SD) unless otherwise stated.

## Tissue staining and histologic analysis

Formaldehyde-fixed paraffin-embedded (FFPE) tissues were prepared using standard methods as described in[77]. Morphologic analysis of tumor tissues was performed on hematoxylin- and eosin-stained sections. For antibody staining, sections were blocked (Dual Endogenous Enzyme Blocking Reagent, Dako, S200389-2) in 1% (w/v) BSA for 60 min at RT, incubated with primary antibodies (Anti-EGFR, D38B1, CST 4267; RRID: AB_2246311) at 4 °C overnight and secondary antibody (2 μg/mL Anti-Rabbit Ig-HRP, Dako P044801 in Tris Borate Saline - TBS) for 60 min at RT. Samples were developed (using Liquid DAB+ Substrate Chromogen System, Dako, K3467) and counterstained with hematoxylin before mounting. Slides were scanned using a Nanozoomer (Hamamatsu, Japan) with images being analyzed and processed by ImageJ.

## Reporting summary

Further information on research design is available in the Nature Portfolio Reporting Summary linked to this article.

## Data availability

FLImP and single particle tracking data generated in this study are available within this paper and upon request from the Lead Contact. Source data are provided with this paper. The confocal data generated in this study to assess binding affinities have been deposited in the Zenodo database under accession code https://doi.org/10.5281/zenodo.10567248[78] and the input files for the MD simulations are deposited in YARETA under accession code https://doi.org/10.26037/yareta:qtkuoibmhndc3jxcwtwzo7eeey[79]. Source data are provided with this paper.

## Code availability

The code used in this study is available upon request from the Lead Contact.

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

## Acknowledgements
We thank Drs Esther Garcia-Gonzalez, Jana Harizanova, Christopher Tynan, Michael Hirsch, Jianguo Rao and Michalis Vrettas for technical support. We thank James Rosekilly and Dr Cheryl Gillett for providing the human tissue TMA that consisted of no/low/high human cancer tissues. We thank Prof Andrew Clayton for his support in collecting the pbICS data. We also thank Dr Michael Hirsch for comments on the manuscript. This work has been funded by grant Ref: ST/S000682/1 from the Science and Technology Facilities Council UK (R.S.I. and B.M.D.) and a joint-funded King's College London-The University of Hong Kong PhD studentship (R.C.H.M.). M.L.M.-F., D.J.R., and B.M.D. are grateful for significant computing resources and support provided by STFC Scientific Computing Department's SCARF cluster and its Data Services, Research Infrastructure and Cloud Operations Groups, with funding from STFC's Ada Lovelace Centre and IRIS eInfrastructure consortium. F.L.G. and I.G. acknowledge the Swiss National Science Foundation and Bridge for financial support (project number: 200021_204795 and 40B2-0_203628). F.L.G. and I.G. also acknowledge the Swiss National Supercomputing Centre (CSCS) for large supercomputer time allocations, project IDs: s1107, s1169, s1228.

## Author contributions
R.S.I., S.R.N., I.G., B.M.D., and S.K.R. contributed equally to this work. Conceptualization, R.S.I., I.G., S.K.R., S.R.N., L.C.Z.-D., P.J.P., D.J.R., B.M.D., D.T.C., F.L.G., and M.L.M.-F.; molecular biology, R.S.I. and S.K.R.; in vitro data acquisition, R.S.I., S.R.N., S.K.R., L.C.Z.-D.; FLImP automation, D.J.R., B.M.D., S.R.N.; algorithm development and data analysis, D.J.R., B.M.D.; MD simulations, I.G. and F.L.G.; in vivo work R.C.H.M., G.O.F., R.S.I., and S.K.R.; writing – original draft, M.L.M.-F.; writing – review & editing, I.G., L.C.Z.-D., R.S.I., S.R.N., D.J.R., B.M.D., G.O.F., P.J.P., S.K.R., D.T.C., F.L.G., M.L.M.-F.; visualization, S.R.N., L.C.Z.-D., B.M.D., D.J.R., S.K.R., D.T.C., I.G., R.C.H.M., R.S.I.; funding acquisition M.L.M.-F., D.J.R., B.M.D., and F.L.G.

## Competing interests
The authors declare no competing interests.

## Additional information

[1]Central Laser Facility, UKRI-STFC Rutherford Appleton Laboratory, Didcot, Oxfordshire, UK. [2]School of Pharmaceutical Sciences, University of Geneva, Geneva, Switzerland. [3]ISPSO, University of Geneva, Geneva, Switzerland. [4]Imaging Therapies and Cancer Group, Comprehensive Cancer Centre, School of Cancer and Pharmaceutical Sciences, Guy's Campus, King's College London, London, UK. [5]Protein Phosphorylation Laboratory, The Francis Crick Institute, London, UK. [6]School of Cancer and Pharmaceutical Sciences, Guy's Campus, King's College London, London, UK. [7]Chemistry Department, University College London, London, UK. [8]Swiss Institute of Bioinformatics, University of Geneva, Geneva, Switzerland. [9]Present address: Immunocore Limited, 92 Park Drive, Milton Park, Abingdon, UK. [10]These authors contributed equally: R. Sumanth Iyer, Sarah R. Needham, Ioannis Galdadas, Benjamin M. Davis, Selene K. Roberts. ✉e-mail: daniel.rolfe@stfc.ac.uk; Francesco.Gervasio@unige.ch; marisa.martin-fernandez@stfc.ac.uk

