## [Peer Review File · Nature Communications]

Drug-resistant EGFR mutations promote lung cancer by stabilizing interfaces in ligand-free kinase active EGFR oligomersEditorial Note: Parts of this Peer Review File have been redacted as indicated to remove third-party material where no permission to publish was obtained.

REVIEWER COMMENTS

Reviewer #1 (Remarks to the Author):

The paper by Iyer and coworkers is a very interesting and well-written study of how the important oncogene EGFR achieves ligand-independent phosphorylation by assembling dimer conformers into obligate ligand-free hetero-conformational oligomers.

They combine bayesian modeling methods and highly advanced spectroscopy to guide the build up of models of oligomers. Importantly, the resolution available for the modeling is twofold, which allows to obtain more accurate constraints and a very precise modeling of the dimer.

All the computational part is extremely well executed and elaborated upon. Classical simulations fully support experiments, indicating that when the interaction surface is stable, MD is able to keep the assembly together, while when a non-stable interface is considered, MD shows instability and break up of the modeled assembly.

It is clear that the comp part has been carried out very well and competently and provides a great support to the experiments.

I liked reading this paper where an interesting combination of experiments and simulations is put in place. As a consequence, I fully support publication of this manuscript.

Reviewer #2 (Remarks to the Author):

In this manuscript, Sumanth Iyer and colleagues investigated the structural details of signaling-competent ligand-free heteroconf-oligomerization within the context of epidermal growth factor receptor (EGFR) signaling. Given its frequent mutations in non-small cell lung cancer (NSCLC), the EGFR serves as a focal point for tyrosine kinase inhibitors, although the emergence of drug resistance poses a significant challenge over time. The authors propose that surmounting this therapeutic hurdle has been impeded by an insufficient grasp of how the wild-type EGFR initiates ligand-independent signals crucial for cellular homeostasis, which are subsequently exploited by oncogenic mutations. And the ligand-independent signaling cannot be explained by the autoinhibited ligand-free EGFR dimer and oligomer structures so far available. By combining super-resolution Fluorophore Localization Imaging with Photobleaching (FLImP), guided by protein structures and MD simulations, and mutagenesis, the authors have unveiled a complex network of previously hidden molecular interactions that drive receptor activation, oligomer structure, assembly mechanisms, and their use by drug resistant EGFR mutants.

The authors first measure oligomer size and discriminate different sub-unit conformers using a higher-resolution FLImP version as a molecular ruler. They then identify specific subunits within heteroconf-oligomers, such as the B2Bect/H2Hkin, the St2Stect/Asymkin and H2Hect/2xkin. They further demonstrate that both wild-type EGFR and oncogenic mutants share a common ligand-free heteroconf-oligomer structure. This structure consists of tetramer scaffolds composed of H2Hect/2xkin monomer and B2Bect/H2Hkin dimer sub-units held by a transversal transmembrane interface cantilever into position the extracellular portion of the St2Stect/Asymkin dimer under the regulation of the ectodomain tethered conformation. The integration of high-resolution imaging with <3 nm precision allowed the authors to delve into the intricate details of ligand-free dimer conformations. These findings offer insights for designing drugs against EGFR mutations by disrupting oligomer formation.

Minor comments:

1. Authors should incorporate a table within the manuscript containing information about all dimers, their separation distances, and the influence of cancer mutations on these distances.
2. Authors should include wild type in Figure 6I.
3. Authors should provide quantification for the western blot bands in figure 6D and I.
4. Authors should provide quantification for the western blot bands in Supplementary Figures 1E, 3E, and 8G.

Reviewer #3 (Remarks to the Author):

In this study, the authors use a fluorescence technique called FLImP that acts as a “molecular ruler”. Based on FLImP measurements of EGFR, the authors draw conclusions about the oligomer size of EGFR and different EGFR mutants. The authors have been using this technique for a while. There are previously published studies which the authors have interpreted as an indication that EGFR forms oligomers in the absence of ligands.

My first reaction while looking at this work is that there are many other studies on EGFR in the absence of ligand, but they do not report a significant abundance of oligomers in the absence of ligand. One example for instance is this paper: Kim, D.-H., Park, S., Kim, D.-K., Jeong, M.G., Noh, J., Kwon, Y., Zhou, K., Lee, N.K., Ryu, S.H. (2018). Direct visualization of single-molecule membrane protein interactions in living cells. *PLoS Biol.* 16(12): e2006660. Another paper is: Stoneman, M. R., Biener, G., Ward, R. J., Padiani, J. D., Badu, D., Eis, A., Popa, I., Milligan, G. and Raicu, V. (2019) A general method to quantify ligand-driven oligomerization from fluorescence-based images. *Nature Methods*, 16(6), pp. 493-496. These are techniques where fixing is not required, and they show that oligomers populations are very small in the absence of ligand. The latter paper describes a technique that is super easy to use, as it just uses a standard confocal microscope. It shows that the major species are EGFR monomers and EGFR dimers in the absence of ligand. As compared to this simple method, the method that the authors use uses complicated math and modeling, and requires fixed cells. So I cannot but worry that what they are seeing is artifact. Can the authors determine what fraction of the EGFR molecules form oligomers in the absence of ligand? Is it 30% or is it 0.3%? Is the percentage changed upon fixing? Is the same percentage observed with other simpler techniques that can give information about the oligomer size (but of course not about the distances that the authors are measuring with FLImP). I agree that the mutations that the authors have engineered affect the tumor growth. But it is not clear if this happens through the mechanism the authors are proposing.

The authors discuss a higher resolution version of FLImP. Is this the first time the method is used? If yes, should it be verified with a known membrane protein oligomer?

A critical question here is whether the oligomeric population of the mutant (T766M) is significant. Again, is it 30% or is it 0.3%. I am sorry if I missed this information, but to me this is the critical question to answer first.

Reviewer #4 (Remarks to the Author):

The authors in this manuscript develop a compressive method, combining experiment with theory, for the investigation of EGFR. Part of the novel contributions in this manuscript is the development of appropriate data analysis methods for at least two different tasks: pre-processing of raw microscopy images and interpretation of the data resulting after pre-processing. For the two tasks, the authors put forward two separate analysis pipelines that combine heuristic and rigorous steps. Unfortunately, the main part of the manuscript is poorly structured and, in its current form, remains incomprehensible. Critical details on what consists the raw data or how FLImP proceeds to collect them are missing. This makes the manuscript inaccessible outside a very narrow group of specialists

already familiar with FLImP and the associated problems that the two pipelines are meant to solve. Due to the poor explanations, a deeper assessment of the technical descriptions in the supplementary methods cannot be given now.

I believe the manuscript will be improved if the authors consider the following remarks.

(1) Big portions of the manuscript need to be rewritten. Especially the introductory paragraphs (lines 65-88 and lines 102-109) need to provide a cleaner picture of the driving principles so uninitiated readers can grasp the setup and appreciate better the challenges the authors attempt to address in supplementary methods 2 and 3. For instance, what is the starting point? What are the quantities of interest? Are these random variables? If so, due to measurement noise, stochastic dynamics, or inefficiency of the analysis methods? What are the causal relationships between them? This last question needs to be clearly answered before the description of the Bayesian analysis (line 556) because it hints upon the validity and construction of the likelihood in lines 560 and 585.

(2) Both pipelines described in supplementary methods 2 and 3 combine many manually set choices and thresholds. For instance, "top 5% quantile", "fewer than 20 frames", "at least 200 tracks lasting at least 80% of duration" and so on. Although it is unreasonable to expect definite documentation of each one of them, an investigation of the consequences is required. For instance, how different would the results be provided if the top 10% quantile is chosen instead of the top 5%?

(3) In addition, the authors combine fundamentally non-Bayesian methods (such as the bootstrap) with Bayesian ones (such as posterior distributions) to establish their data deconvolution. Such a combination is not theoretically possible. My impression is that for the deconvolution the authors developed a theoretical method within the ABC (approximate Bayesian computation) family rather than a fully Bayesian one, although their description does not really support this viewpoint. To help with clarity, I believe the authors should expand their description in lines 556-566 and either adapt the terminology appropriate in ABC or more clearly present how bootstrap results model their posterior as opposed to the natural one of how bootstrap may model their likelihood.

(4) Finally, the authors should consider a comparison of their 1D deconvolution method with a much simpler Bayesian or non-Bayesian clustering algorithm, for instance K-means, non-Gaussian mixture models, or simpler deconvolution techniques. Especially, a comparison with a non-parametric clustering algorithm, such as an infinite mixture model, could help clarify if model selection (lines 687-697) is a critical requirement or not.

Responses to Reviewer 1

We thank the reviewer for their favorable comments on the clarity of the manuscript and the quality of the work. We are grateful for the appreciation of the efforts that it entailed.

Responses to Reviewer 2

We thank the reviewer for their insightful comments and address below the points raised:

Authors should incorporate a table within the manuscript containing information about all dimers, their separation distances, and the influence of cancer mutations on these distances.

We agree with the reviewer that a summary table should be of help to the readers and have included the requested table in Supplementary information (**Supplementary Table 1**). We constructed the table as follows: (i) We first calculated the separations predicted by the model; (ii) We then examined how these separations are satisfied by the different conditions and annotated the presence or absence for each one; (iii) We highlighted which separations depend on each of the three ligand-free dimers and marked the changes introduced by the cancer mutations.

Authors should include wild type in Figure 6I.

The blot including wild type requested by the reviewer was already included in the submitted manuscript, but we had placed it in supplementary info (see old Supplementary Fig. 8G in the first submitted version of the manuscript). Instead of the blot without wild type shown before, we now refer to these WB data containing the wild type band (see line 330 main text). We note that we have had to rearrange the supplementary figures to comply with formatting issues; thus, the reviewer can find the blot required and its quantification in **Supplementary Fig. 7c** in the revised manuscript.

Authors should provide quantification for the western blot bands in figure 6D and I.

As mentioned in the point above, the blot of Fig. 6I in the original submitted version of the manuscript has been replaced with the blot in **Supplementary Fig. 7c** in the revised version, which now includes quantification. We note that we had to include an additional figure in the revised main text (new **Fig. 2**), which required breaking Fig. 1 in two to comply with formatting. For this reason, the blot in Fig. 6D is now in **Fig. 8d** in the revised version. As requested by the reviewer, we have included quantification of this blot in **Supplementary Fig. 7b**. See also revised main text, lines 330, 428, and 1048.

Authors should provide quantification for the western blot bands in Supplementary Figures 1E, 3E, and 8G.

Quantification of these blots is now in **Supplementary Fig. 1e**, **Supplementary Fig. 3e**, and **Supplementary Fig. 7c**.

Responses to Reviewer 3

We thank the reviewer for their comments and address below the points raised:

There are many other studies on EGFR in the absence of ligand, but they do not report a significant abundance of oligomers in the absence of ligand ...

The implication here is that the absence of ligand-free oligomers is an established fact, but this is not the case. In our own published work, e.g.:

- Zanetti-Domingues et al, Nat Comm 2018, <https://doi.org/10.1038/s41467-018-06632-0>
- Needham et al, Nat Comms 2016, <https://doi.org/10.1038/ncomms13307>
- Needham et al, PloS one 2013, <https://doi.org/10.1371/journal.pone.0062331>
- Needham et al, Bioch. Soc. Trans. 2014, <https://doi.org/10.1042/BST20130236>

- Zanetti-Domingues et al, Prog. Biophys. Mol. Biol. 2015, <https://doi.org/10.1016/j.pbiomolbio.2015.04.002>

we show the presence of ligand-free oligomers not only using FLImP, but also by other methods and in live cells, e.g., counting the photobleaching steps of moving particles using single particle tracking. See for example (https://static-content.springer.com/esm/art%3A10.1038%2Fncmms13307/MediaObjects/41467_2016_BFncmms13307_MOESM1033_ESM.pdf).

There are also other published studies in live cells that use various methods that report oligomerization in the absence of ligand, for example:

- Saffarian et al, Biophys J. 2007, doi: 10.1529/biophysj.107.105494
- Wollman et al, J. R. Soc. Interface 2022, <https://doi.org/10.1098/rsif.2022.0088>
- Byrne et al, J Biol Chem 2020, <https://doi.org/10.1098/rsif.2022.0088>
- Balasubramanian et al, 2022, *Biophys J*. DOI:<https://doi.org/10.1016/j.bpj.2022.11.003>

... *One example for instance is PLoS Biol. 16(12): e2006660*

We are familiar with the work referenced by the reviewer. From our understanding, the aim of that work is to determine the dynamics, specificity, and strength of interactions by combining single particle tracking and immuno-immobilization. We are unsure of which point the Reviewer is trying to make regarding this paper because as far as we can tell this work does not report stoichiometric fractions. Using these methods, to our knowledge, measuring the stoichiometry of the moving particles would not have been possible because the authors photobleached cells to an image spot density at which only individual fluorophores were detected. On page 5 of the article, second paragraph, the authors write: "*We additionally corrected the total number of SNAP-EGFRs, considering the proportion of nonfluorescent CF660R, which should be immobilized but not detected*". We apologize if we have missed something.

... *Another paper is: Nature Methods, 16(6), pp. 493-496. These are techniques where fixing is not required, and they show that oligomers populations are very small in the absence of ligand. The latter paper describes a technique that is super easy to use, as it just uses a standard confocal microscope. It shows that the major species are EGFR monomers and EGFR dimers in the absence of ligand.*

In this example suggested by the reviewer (Nat Methods 2019), the confocal-based method cited, fluorescence intensity fluctuation spectrometry (FIF), report a distinguishable population ligand-free oligomers for wild type EGFR (see on the left highlighted Fig. 2c [redacted]) and for an EGFR mutant (supplementary fig. 3 from the cited Nat Methods 2019).

We could not ascertain in the paper whether these results had been corrected for probe photobleaching. If they have not, we would like to offer a possible explanation why the fraction of oligomers may be lower in this example than that measured by other

methods. In our experience using confocal systems, at the power density used in the Nat Methods paper cited by the reviewer (average power per voxel of 3.9 mW), almost immediately after opening the shutter one would visibly lose >20% of the fluorophores by photobleaching. From there on, exponential bleaching continues. In our hands, one needs to be very careful to preserve half of the fluorophores on average whilst measurements are taken.

In addition, it is unfeasible to label all molecules of interest. Using fluorescent proteins (FPs), among the most optimal labels, and those used in the two examples cited by the reviewer (PLoS Biol. 2018 and Nat Methods 2019), in the best cases only up to 80% of FPs become fluorescent (Ulbrich et al, Nat Methods 2007, doi:10.1038/NMETH1024).

Let's assume the best fraction that can be labeled (80%) and favorable conditions in which 50% are not bleached during our measurements. Our fluorescent fraction (f) would be $f = 0.8 \times 0.5 = 0.4$. Using a binomial distribution, which assumes each labeling site within each structure is independent, with N = number of real binding sites, and M = the number the number of sites detected, there will be substantial differences between what one detects, and the underlying truth as shown in the Fig. below.

In the hypothetical example above (red bars), we permit oligomers up to octamers with an abundance consistent with our previous work (<https://doi.org/10.1038/s41467-018-06632-0>). For $f = 0.4$, experimental results (blue bars) would artifactually suggest that higher order oligomers are substantially less prevalent than they indeed are. Differences will get even worse as f decreases further, which could possibly explain why methods using lower power densities might report higher fractions of oligomers than other methods that employ relatively higher power densities (e.g. confocal), unless this is taken into account.

Whilst sub-optimal labelling and photobleaching are among possible explanations why the fraction of ligand-free oligomers one measures may vary between techniques, the important point we would like to convey is that different techniques have nevertheless managed to detect ligand-free oligomers. Thanks to this, we can finally understand how EGFR can become phosphorylated in the absence of ligand, a phenomenon that is otherwise inexplicable in the context of monomers and autoinhibited ligand-free dimers, given that ligand-free active dimers have not got enough free-energy to form by themselves.

To understand the mechanisms of EGFR phosphorylation in the absence of bound ligand is not only essential to understand EGFR-dependent cell homeostasis, but also EGFR-dependent cell transformation, which depends on the dysregulation of ligand-free EGFR phosphorylation mechanisms, i.e. oligomerization. The advance we make in the submitted manuscript is that we reveal the mechanism by which ligand-free oligomers achieve phosphorylation, and how cancer mutations exploit these ligand-free oligomerization mechanisms to transform cells.

The method that the authors use uses complicated math and modeling ...

The math and modeling are required to eliminate limitations from other methods and achieve high resolution. Because of these math and modeling,

- FLImP does not require baseline standards.
- FLImP does not depend on the fraction of labeling.
- FLImP allows you to obtain <3 nm resolution.

Resolution is critical for our work aims because diffraction-limited techniques, like confocal, can only infer whether molecules are co-localized within $\sim >250$ nm. We want to determine if our molecules of interest (~ 10 nm in size) interact to form stoichiometric complexes and determine their structure. FLImP uniquely reports the fingerprints of molecular interactions, hence it allows structural determination and a mechanistic understanding of these interactions in cells.

... and requires fixed cells. So I cannot but worry that what they are seeing is artifact.

We have used FLImP since 2008 and have worked intensely to reassure ourselves our results are sound and robust. Some of the controls for the fixation procedure include: In Needham et al 2016, <https://doi.org/10.1038/ncomms13307> (Supplementary Figure 11), we crucially showed that the brightness in single particle spots is the same before and after fixation, which strongly argues that the fixative has not induced aggregation. We additionally carried out the following controls to further validate our results: In <https://doi.org/10.1038/ncomms13307>, we showed that high EGF concentrations (~ 100 nM) and mutations disrupt oligomerisation. In another control, we showed that receptors move together in groups in live cells in the absence of bound ligands. In doi: 10.1042/BST20140318, we carried out consistency tests that showed that we could reproduce the expected breaking of higher-order EGFR oligomers into dimers upon receptor downregulation by phorbol myristate acetate (PMA) activation of protein kinase C (PKC) and the expected cancellation of the effect of the PMA treatment by PKC inhibitor bisindolylmaleimide-I (BM-I). In doi: 10.1016/j.ymeth.2015.05.009, we showed that FLImP distributions change as expected upon cholesterol removal from the membrane.

Nevertheless, because it always remains possible in principle that certain artefacts may have escaped our scrutiny and affected our results, in addition, in every paper where we have used FLImP, including the current manuscript, we always additionally test predictions of our FLImP results. For example, the Reviewer can find in Fig. 4e of the original submission and **Fig. 6e** of the revised version a test in which we reproduced results for an intracellular deletion mutant by inhibiting the intracellular interfaces that hold together full-length receptors in oligomers. In this manuscript we have also tested predictions of our FLImP results using single particle tracking in live cells. The Reviewer can find this in a sentence (line 148) pointing to **Supplementary Fig. 1b**, in which single particle imaging tests were carried out aimed at detecting artefacts, and which validate predictions from our results. Indeed, the exquisite agreement between the results from the FLImP data and the predictions from single particle tracking MD simulations, and *in vitro* and *in vivo* functional assays, together with our failure to detect artefacts by other means, convinces us that our data are sound.

Can the authors determine was [sic] fraction of the EGFR molecules form oligomers in the absence of ligand? Is it 30% or 0.3%?

We determined $\sim 30\%$ using EGFR-eGFP and model-free photobleaching image correlation spectroscopy (pbICS). The reviewer can find our results in Fig. 2C of Zanetti-Domingues et al, Nat Comms 2018. DOI: 10.1038/s41467-018-06632-0.

Is the percentage changed upon fixing?

Similar results have been reported using live cells and EGFR-eGFP by Saffarian et al, Biophys J. 2007, doi: 10.1529/biophysj.107.105494, using the FIDA method (a combination of fluorescence correlation spectroscopy and fluorescent brightness analysis). We have also conducted tests in which the brightness of

well-resolved single particle spots does not change upon fixation. Needham et al, Nat Comms 2016, <https://doi.org/10.1038/ncomms13307>.

Is the same percentage observed with other simpler techniques that can give information about the oligomer size

The methods that arrive at >30% ligand-free EGFR oligomer fraction include model-free photobleaching image correlation spectroscopy (fixed cells), FIDA (live cells), and TIRF + brightness measurements (live cells) (refs in our response to the first point above).

I agree that the mutations that the authors have engineered affect the tumor growth. But it is not clear if this happens through the mechanism the authors are proposing.

We carried out the work in BA/F3 cells and mice to test predictions from our FLImP results. In this work, we disrupted oligomerization by the H566F mutation, which inhibits the tether conformation, and disrupted oligomerization and phosphorylation together via the K946E mutation, which inhibits the Bb2Bb^{kin}_{dimer}. These predictions are counterintuitive and could not have been derived from any previous data. The excellent agreement throughout testifies to the robustness of our method.

The authors discuss a higher resolution version of FLImP. Is this the first time the method is used? If yes, should it be verified with a known membrane protein oligomer?

We thank the reviewer for the suggestion. We have collected data from CHO cells expressing wild type EGFR with 4 nM EGF-CF640R and analyzed these data using our enhanced resolution analysis. We have reproduced the positions of the peaks we published in Nat Comms 2016, but please note the >2-fold increase in resolution (1.7 nm below v 4.8 nm in Nat Comms 2016). This figure has been added to Supplementary information (**Supplementary Fig. 1a**).

A critical question here is whether the oligomeric population of the mutant (T766M) is significant. Again, is it 30% or is it 0.3%. I am sorry if I missed this information, but to me this is the critical question to answer first.

In previous work (unpublished) we found using pbICS that the oligomeric population of WT EGFR in the absence of ligand is ~30%. We have also used this well-established technique to measure the oligomeric populations of other EGFR mutants, as shown below:

Mutant	% of oligomers
K721A	26.7
C'698	20.3
ΔC	43.7
L680N	39.3

The data show a high population of oligomers for all the EGFR mutants we measured, and we cannot think of any reason that the T766M mutation will introduce a specific mutation-dependent artifact. Besides, there is excellent agreement between the predictions from T766M FLImP results and MD simulations published in Nat Comms 2018.

Responses to Reviewer 4

We thank the reviewer for their efforts.

General remarks: We were initially a bit surprised by the comments from this Reviewer until, on inspection of the submitted manuscript, we realized that many critical explanations of the FLImP methods were missing. In the process of moving some method descriptions from the main text to the supplementary information due to the word limit, several sections were somehow inadvertently dropped, and we failed to notice. We

can only apologize for our unintended mistake and are truly sorry for the time the reviewer must have wasted trying to make sense of our work without the complete information. We have now rectified this by the addition of the missing methods.

Our responses to the many helpful points raised are below:

Unfortunately, the main part of the manuscript is poorly structured and, in its current form, remains incomprehensible. Critical details on what consists the raw data or how FLImP proceeds to collect them are missing. This makes the manuscript inaccessible outside a very narrow group of specialists already familiar with FLImP and the associated problems that the two pipelines are meant to solve. Big portions of the manuscript need to be rewritten. Especially the introductory paragraphs (lines 65-88 and lines 102-109) need to provide a cleaner picture of the driving principles so uninitiated readers can grasp the setup and appreciate better the challenges the authors attempt to address in supplementary methods 2 and 3. For instance, what is the starting point? What are the quantities of interest? Are these random variables? If so, due to measurement noise, stochastic dynamics, or inefficiency of the analysis methods? What are the causal relationships between them? This last question needs to be clearly answered before the description of the Bayesian analysis (line 556) because it hints upon the validity and construction of the likelihood in lines 560 and 585.

Guided by the fair points raised by the reviewer, we have added clarifications to the paragraphs of the main text (lines 64-90) and introduced a new **Fig. 2** to explain the principles of the FLImP method, which are outlined in the caption. We have also revised all the other captions improving clarity where possible, and added the references below for completeness (refs 34-38 in line 73).

<https://doi.org/10.1038/s41467-018-06632-0>; <https://doi.org/10.1038/ncomms13307>
<https://doi.org/10.1016/j.pbiomolbio.2015.04.002>; <https://doi.org/10.1016/j.ymeth.2015.05.009>
<https://doi.org/10.1042/BST20140318>; <https://doi.org/10.1371/journal.pone.0062331>

In Supplementary Method 3 (FLImP analysis) we have also included a summary overview of the method (section I, page 27) in which we have listed each stage in the analysis process before the detailed description of these methods (section II, page 28). Most importantly, in Supplementary Method 3 we have added the sections that were unintentionally missing (section 1, 4, 5, 7, 8 and 9) (see supplementary information pages 27, 31, 38, 39).

Both pipelines described in supplementary methods 2 and 3 combine many manually set choices and thresholds. For instance, "top 5% quantile", "fewer than 20 frames", "at least 200 tracks lasting at least 80% of duration" and so on. Although it is unreasonable to expect definite documentation of each one of them, an investigation of the consequences is required. For instance, how different would the results be provided if the top 10% quantile is chosen instead of the top 5%?

We describe and apply here a set of algorithms to reimplement and refine in an automated fashion our established processes (see the refs in our response above) to acquire, identify and analyze tracks which are consistent with the assumptions of FLImP localization fit. This initially partly manual and now fully automated process has been in use and carefully refined by the same team over more than a decade and has always focused on identifying the data and applying analyses which follow the same set of assumptions. Over this whole period the entire process has continued to reproduce the same results for the same conditions, apart from improving resolution. This is exemplified in the figure below [redacted]. Both data sets shown were collected from the same sample type (CHO cells stably expressing wild type EGFR) and labelled in the same way (4 nm EGF-CF640R). The data set at the top was collected using manual data acquisition procedures and the data published in Fig. 1c (DOI: 10.1038/ncomms13307). The one at the bottom was collected using the automated

procedure designed to mimic its manual predecessor. Note the overall consistency between the two data sets. The data in the top uses a Rician fit, and the data in the bottom achieves better resolution by means of the Bayesian procedures we have developed.

It is important to note that the parameters of the final automated approach described here were all fixed a-priori to achieve and refine in our judgement the same process as previously manually performed. This was done before acquisition of any of the data presented in this paper, with the parameters then remaining fixed for all data in the paper, and reproduced results consistent with our previously published work. We have added a comment to this effect to the new FLImP analysis process overview section in the Supplementary Methods 3.

In addition, the authors combine fundamentally non-Bayesian methods (such as the bootstrap) with Bayesian ones (such as posterior distributions) to establish their data deconvolution. Such a combination is not theoretically possible. My impression is that for the deconvolution the authors developed a theoretical method within the ABC (approximate Bayesian computation) family rather than a fully Bayesian one, although their description does not really support this viewpoint. To help with clarity, I believe the authors should expand their description in lines 556-566 and either adapt the terminology appropriate in ABC or more clearly present how bootstrap results model their posterior as opposed to the natural one of how bootstrap may model their likelihood.

We suspect this is a misunderstanding due to the missing “Posterior comparisons and bootstrap resampling” (section 7 in Supplementary Method 3, overview (page 27) and detailed description (page 38)). This is not an approximate Bayesian computation. We perform a Bayesian MCMC sampling from a posterior to decompose our finite sample of independent separation measurements into a discrete set of precise components. We then perform bootstrap resampling of our set of measurements, repeating the Bayesian analysis for each resampled dataset to consider the effect of a finite set of measurements when comparing the decomposition results between multiple conditions.

Finally, the authors should consider a comparison of their 1D deconvolution method with a much simpler Bayesian or non-Bayesian clustering algorithm, for instance K-means, non-Gaussian mixture models, or simpler deconvolution techniques. Especially, a comparison with a non-parametric clustering algorithm, such as an infinite mixture model, could help clarify if model selection (lines 687-697) is a critical requirement or not.

Two of our previous FLImP papers (Zanetti-Domingues et al. (2018). Nat. Commun. 9, 4325. 10.1038/s41467-018-06632-0 and Needham, S.R. et al. (2016). Nat. Commun. 7, 13307. 10.1038/ncomms13307) did apply a Rician-mixture modelling approach to the sum (effectively a histogram) of the individual separation measurement posteriors. This method works quite well but is sub-optimal as by summing the measurement posteriors first, it loses the ability to consider the hypothesis separately for each measurement posterior, and so cannot fully take advantage of the robust uncertainties we have determined in each measurement, for example, to distinguish between a cluster of closely spaced but robustly resolved measurements and a single broad component of imprecise, unresolvable measurements. Our improved method was conceived to address this by forming a posterior for the model, which considers all measurements individually.

Techniques such as K-means typically assume equal abundance and variance of each separation and with no accounting for spurious detections (clutter) and would therefore be expected to perform poorly with this dataset.

Especially, a comparison with a non-parametric clustering algorithm, such as an infinite mixture model, could help clarify if model selection (lines 687-697) is a critical requirement or not.

The model selection problem is definitely a challenging one, still considered an unsolved problem in statistics. Although much progress has been made there is no definitive ‘right answer’ yet. We believe using the simplest model demanded by the data using the BIC is a justified approach for this work, and it yields convincing results, for example as demonstrated in the simulations shown in Fig 1d, 1e, 1g, 1h. However, the choice between choosing a single justified model as we do now and sampling probabilistically from all models and using them all accordingly is something we are already considering for future improvements. The referee’s comments led us to discover this paper (Rasmussen, C.E. ‘The Infinite Gaussian Mixture Model’, in *Advances in Neural Information Processing Systems 12* S.A. Solla, T.K. Leen and K.-R. Muller (eds.), pp. 554–560, MIT Press (2000)) of which we were not previously aware. That work fits an infinite mixture model, and has some similarities to our approach, particularly that it considers the ambiguity in assignment between measurements and model components in a very similar way to our method, and that it uses an MCMC sampling method to evaluate the posterior. It also samples from all numbers of components to produce a distribution of component locations, marginalizing out the number of components, but this is done by making some assumptions to yield prior information on the distribution and number of components which we wish to avoid. It also does not handle uncertainties in individual measurements, something which, as discussed above, provides powerful information to constrain the problem, and which our method is explicitly designed to exploit. Furthermore, it does not consider the idea of spurious “clutter” measurements. We believe it supports key aspects of our approach, and we will definitely consider further to see how an improved method for future work might incorporate the best of both.

We want to apologize again for our unintended omission in supplementary method 3, which we imagine must have made the task of Reviewer 4 much harder. We also hope that the changes and additions to the manuscript guided by the Reviewers’ efforts are satisfactory.

REVIEWER COMMENTS

Reviewer #2 (Remarks to the Author):

My concerns were addressed.

Reviewer #3 (Remarks to the Author):

I am not very convinced by the rebuttal of the authors, which mainly consists of claims that other researchers do not know how to do experiments. In PLoS Biol. 16(12): e2006660, the authors state: "We analyzed the frequency of stopping EGFR in the vicinity of the immobilized one, which enables us to infer the distribution of the oligomer size (S12 Fig). This stoichiometry analysis implies that EGFR dimer is a major population, with a small portion of oligomers induced by EGF."

I appreciate that the authors have provided information about the fraction of oligomeric receptors in the rebuttal (20 to 30%). They should add this information in the paper, and they should report the EGFR expression level for which this was measured, as well as the expression of EGFR in vivo. This oligomeric fraction will depend strongly on EGFR expression, so the authors should estimate the fraction of oligomeric receptors in vivo.

If the oligomeric fraction of receptors is 20%, then a population of 100 EGFR molecules could be distributed as: 50 monomers, 15 dimers, and 5 tetramers.

Are the 5 tetramers the only signaling molecules? How many signaling EGFR oligomers are present in a cell in vivo? Is there ligand in the in vivo experiments? Are there ligand-free oligomers if there is ligand in vivo?

If the EGFR expression is low, the population of oligomers may be negligible even if the authors "see" oligomers in their fluorescence experiments. Just showing an effect of a mutation in vivo does not necessarily mean that a hypothesized mechanism is correct. Signaling is complex.

Reviewer #4 (Remarks to the Author):

The authors have substantially revised their manuscript and successfully addressed my comments or provided adequate reasons for not doing so. Especially, the addition of Fig 2 and the rewording of the main part of the manuscript have improved the clarity of the methods used. After reviewing the revised manuscript and the expanded documentation, I am happy to report that the manuscript's methods are appropriate and, especially, the data analysis is carefully implemented and executed.

Responses to Reviewer 3

We thank the reviewer for very helpful and insightful comments, which have helped us to think through the meaning and implications of our *in vivo* results. We believe the manuscript has been improved as a consequence. We address below the points raised (changes made to the manuscript in bold).

I am not very convinced by the rebuttal of the authors, which mainly consists of claims that other researchers do not know how to do experiments ...

It was not our intention to suggest that experiments by others were not properly executed. Our intention was to indicate potential issues that can result from photobleaching in certain types of experiments.

I appreciate that the authors have provided information about the fraction of oligomeric receptors in the rebuttal (20 to 30%). They should add this information in the paper, and they should report the EGFR expression level for which this was measured ...

We have added these data in Supplementary Fig. 1a and caption. Our results indicate that 15-40% of receptors are incorporated into oligomers in CHO cells expressing 10^5 receptor copies/cell. For the sake of consistency, in Supplementary Fig. 1a we included the WT-EGFR experiment that accompanied the same experimental run in which the mutants were investigated, and not the result we obtained in a previous effort (Zanetti-Domingues et al, Nature Comms 2018, DOI: 10.1038/s41467-018-06632-0).

We have also added a short paragraph stating expression levels and fraction of receptors incorporated in dimers and oligomers (lines 114-116).

... as well as the expression of EGFR in vivo. This oligomeric fraction will depend strongly on EGFR expression, so the authors should estimate the fraction of oligomeric receptors in vivo.

We have carried out additional immunohistochemistry experiments that indicated high level of expression in the Xenograft tumors in Fig. 10. We estimate $\sim 200,000$ receptor copies/cell, **Supplementary Fig. 10f and caption**). Because the fraction of oligomers depends on receptor expression (Byrne et al, 2020, doi: 10.1074/jbc.RA120.012852) we would not expect the fraction of oligomers to be too dissimilar to the CHO cells.

We have added a paragraph with these results (lines 459-463).

If the oligomeric fraction of receptors is 20%, then a population of 100 EGFR molecules could be distributed as: 50 monomers, 15 dimers, and 5 tetramers.

Starting from the fraction of receptors incorporated in oligomers of 15-40%, we calculated that the fraction of oligomers is $\sim 2-10\%$, depending on oligomer size. We added this in **Supplementary Fig. 1a** and caption.

Are the 5 tetramers the only signaling molecules? How many signaling EGFR oligomers are present in a cell in vivo? Is there ligand in the in vivo experiments?

These are great questions that have helped us to reflect on what the *in vivo* data was telling us. Previous work using intravital multiphoton imaging, confocal imaging, and biochemical analysis showed that the concentration of ligand in tumor xenografts from the human oral squamous carcinoma HSC3 cell line *in vivo* is 17-100 pM (Pinilla-Macua et al, eLife 2017, <https://doi.org/10.7554/eLife.31993.001>). We also know that the concentration of EGFR ligands in human body fluids and tumors is typically <1 ng/ml (<150 picomolar) (e.g., Ishikawa et al. 2005 <https://doi.org/10.1158/0008-5472.CAN-05-1556>; Rich et al. 2017, <https://doi.org/10.2147/CMAR.S115835>; Dvorak et al 2010, <https://doi.org/10.1016/j.jpeds.2009.11.018>).

Pinilla-Macua et al also showed that the small pool of EGFRs that can bind the picomolar concentration of ligand is responsible for driving tumor growth. This was based on functional studies, which reproduced the behavior expected from very low number of receptors (<10,000) binding pM concentrations of EGF. These results were validated in tumor xenografts from other cell lines (NSCLC H322 and triple negative breast cancer MDA-MB-468). Based on the overlap between the concentration of ligand in tumor xenografts and the K_D values measured for high-affinity EGF binding to EGFR in cultured cells (10-100 pM) (Ringerike et al 1998, <https://doi.org/10.1074/jbc.273.27.16639>; Rees et al 1984, <https://doi.org/10.1002/j.1460-2075.1984.tb02057.x>; Sorkin et al, 1991 [https://doi.org/10.1016/S0021-9258\(18\)92983-2](https://doi.org/10.1016/S0021-9258(18)92983-2)), Pinilla-Macua et al assigned the small pool of tumor-driving receptors to the 2-5% of EGFRs that bind EGF with high-affinity (Mattoon et al, 2004, <https://doi.org/10.1073/pnas.0307286101>).

An additional important finding by Pinilla-Macua et al is that the small fraction of receptors occupied by pM concentrations of ligand drive tumor growth via signaling through the Ras-MAPK pathway, mirroring previous results in several types of cultured cells that showed that the small pool of EGFR occupied by pM concentrations of EGF is sufficient to fully activate the ERK1/2 signaling pathway (Albeck et al., 2013, <https://doi.org/10.1016/j.molcel.2012.11.002>; Shi et al., 2016, <https://doi.org/10.1126/scisignal>; Krall et al., 2011, <https://doi.org/10.1371/journal.pone.0015945>).

The work summarized above suggests a possible mechanism by which the mutations we introduced inhibit tumor growth *in vivo*.

- **We have described this in a paragraph in the Discussion (lines 524-539).**

In a nutshell, in the submitted work we found that $\text{St2St}^{\text{ect}}/\text{Asym}^{\text{kin}}_{\text{dimer}}$ sub-units in oligomers are the sites where EGF binds with high-affinity. These constitute 25-30% of the total EGFR binding sites displayed by the oligomers, i.e. ~2-6% of the total binding sites (25-30% of the 1/2 of the 15-40% receptors incorporated in oligomers larger than hexamers, Zanneti-Domingues et al 2018, <https://doi.org/10.1038/s41467-018-06632-0>). We note the smallest oligomer unit that can bear one $\text{St2St}^{\text{ect}}/\text{Asym}^{\text{kin}}_{\text{dimer}}$ is a hexamer (see model Fig. 7). We also found that the $\text{St2St}^{\text{ect}}/\text{Asym}^{\text{kin}}_{\text{dimer}}$ sub-unit is an oligomer obligate because it is inhibited by inhibiting the $\text{Bb2Bb}^{\text{kin}}_{\text{interface}}$, which is required to buttress the $\text{Asym}^{\text{kin}}_{\text{dimer}}$. The two mutations that interfere with tumor growth either eliminate (K946E) or interfere with (H566F) high-affinity EGF-binding sites.

We have additionally made the following changes to the text to highlight existing results in order to set up the paragraph we have added to the Discussion where we outline the possible mechanism by which our mutations inhibit tumor growth:

- **We expanded the rationale that guided us to use ligand binding affinity to distinguish between two dimer conformers (lines 177-181).**
- **A sentence explaining the importance of high-affinity ligand binding (lines 195-196)**
- **A sentence explaining that the H566F mutation makes the ligand binding site less accessible (lines 252-254). This information was already in supplementary Fig. 2 and caption, but it was needed in the main text to set up the discussion.**
- **A sentence explaining that lapatinib experiments support our assignment of high-affinity sites (lines 273-274)**
- **A sentence explaining that our model predicts that a hexamer would be the smaller unit (lines 340-342)**
- **A sentence explaining that T766M-EGFR oligomers stabilized via strengthening of the $\text{H2H}^{\text{kin}}_{\text{dimer}}$ bear more $\text{St2St}^{\text{ect}}/\text{Asym}^{\text{kin}}_{\text{dimer}}$ sub-units, explaining the increase in ligand-independent phosphorylation (lines 418-420).**
- **A small amount of text tidying (lines 433-436)**

Are there ligand-free oligomers if there is ligand in vivo?

It is known that >~50% of cells in the tumor (HSC3) are accessible to circulating ligands (Pinilla-Macua et al). EGFR can trigger signals for growth and division when cells are exposed to ligand (Chakraborty et al, 2014 DOI: 10.1038/ncomms6811). The T766M mutation would therefore drive tumor growth by stabilizing high-affinity binding sites within larger oligomers. The high-affinity binding sites of oligomers of T766M-EGFR in growing cells would be occupied by pM ligand concentrations.

Oligomers formed by T766M+K946E-EGFR, which do not display high-affinity binding sites, would not be occupied by ligand at the pM concentrations available. Because of this, cells expressing T766M+K946E-EGFR do not grow, explaining why tumors don't develop in the presence of this mutation.

Beyond the revisions described above (bold), we have made the following small changes:

Lines 54-55: T766M is also commonly known as T790M, and we state this to help some audiences.

Line 66: We deleted a paragraph where information was repeated.

Lastly, we would like to thank the reviewer again for a great review and for guiding us to understand better the in vivo data.

REVIEWERS' COMMENTS

Reviewer #3 (Remarks to the Author):

I have read the revised paper. The authors will have to think what to put in the abstract, as the current version does not make it clear what the take-home message is. This is a complex paper, and the abstract ought to help the readers. I am still confused, is the unliganded EGFR oligomer important physiologically? Am I correct to think that the new revisions say that it is not, that ligand is critical? That would be consistent with papers saying that in the absence of ligand EGFR cannot trigger downstream signaling. Am I understanding correctly? Perhaps it should be "drive" instead of "driving" on line 499.

Responses to Reviewer 3

We thank again the reviewer for the comments. We address below the points raised (all changes made to the manuscript have been tracked).

I have read the revised paper. The authors will have to think what to put in the abstract, as the current version does not make it clear what the take-home message is. This is a complex paper, and the abstract ought to help the readers.

We have revised the abstract to clarify aspects that might have been confusing as suggested.

I am still confused, is the unliganded EGFR oligomer important physiologically?

The unliganded oligomer is critical. We describe the mechanisms by which ligand-free oligomers assemble from two autoinhibited dimer conformers, and how these chaperone the formation of oligomer-obligate active sub-units bypassing the need for ligand. If the interfaces in ligand-free oligomers by which active units can form are inhibited, tumours do not grow. The key link is that these active sub-units are also those that bind ligand with high affinity, on which the growth of tumors depend.

Am I correct to think that the new revisions say that it is not, that ligand is critical? That would be consistent with papers saying that in the absence of ligand EGFR cannot trigger downstream signaling. Am I understanding correctly?

Ligand is critical for canonical signaling that triggers tumor growth. For physiological pM concentrations of ligand to bind, integral ligand-free oligomers capable of auto-phosphorylation, which are those that contain the high-affinity sites, are needed. If this is inhibited, pM ligand concentrations cannot bind and tumors cannot grow. We have made this explicit in this revised version. We are grateful to the reviewer for helping us to be clear about this.